# ONE TRANSFORMER CAN UNDERSTAND BOTH 2D & 3D MOLECULAR DATA

**Shengjie Luo**[1],      **Tianlang Chen**[2,5]*,      **Yixian Xu**[2]*,
**Shuxin Zheng**[3],      **Tie-Yan Liu**[3],      **Liwei Wang**[1,4]†,      **Di He**[1]†
[1]National Key Laboratory of General Artificial Intelligence, School of Intelligence Science and Technology, Peking University   [2]School of EECS, Peking University   [3]Microsoft Research
[4]Center for Data Science, Peking University        [5]Shanghai Artificial Intelligence Laboratory
`luosj@stu.pku.edu.cn, tlchen@pku.edu.cn, xyx050@stu.pku.edu.cn`
`{shuz, tyliu}@microsoft.com, {wanglw, dihe}@pku.edu.cn`

## ABSTRACT

Unlike vision and language data which usually has a unique format, molecules can naturally be characterized using different chemical formulations. One can view a molecule as a 2D graph or define it as a collection of atoms located in a 3D space. For molecular representation learning, most previous works designed neural networks only for a particular data format, making the learned models likely to fail for other data formats. We believe a general-purpose neural network model for chemistry should be able to handle molecular tasks across data modalities. To achieve this goal, in this work, we develop a novel Transformer-based Molecular model called Transformer-M, which can take molecular data of 2D or 3D formats as input and generate meaningful semantic representations. Using the standard Transformer as the backbone architecture, Transformer-M develops two separated channels to encode 2D and 3D structural information and incorporate them with the atom features in the network modules. When the input data is in a particular format, the corresponding channel will be activated, and the other will be disabled. By training on 2D and 3D molecular data with properly designed supervised signals, Transformer-M automatically learns to leverage knowledge from different data modalities and correctly capture the representations. We conducted extensive experiments for Transformer-M. All empirical results show that Transformer-M can simultaneously achieve strong performance on 2D and 3D tasks, suggesting its broad applicability. The code and models will be made publicly available at `https://github.com/lsj2408/Transformer-M`.

## 1 INTRODUCTION

Deep learning approaches have revolutionized many domains, including computer vision (He et al., 2016), natural language processing (Devlin et al., 2019; Brown et al., 2020), and games (Mnih et al., 2013; Silver et al., 2016). Recently, researchers have started investigating whether the power of neural networks could help solve important scientific problems in chemistry, e.g., predicting the property of molecules and simulating the molecular dynamics from large-scale training data (Hu et al., 2020a; 2021; Zhang et al., 2018; Chanussot et al., 2020).

One key difference between chemistry and conventional domains such as vision and language is the multimodality of data. In vision and language, a data instance is usually characterized in a particular form. For example, an image is defined as RGB values in a pixel grid, while a sentence is defined as tokens in a sequence. In contrast, molecules naturally have different chemical formulations. A molecule can be represented as a sequence (Weininger, 1988), a 2D graph (Wiswesser, 1985), or a collection of atoms located in a 3D space. 2D and 3D structures are the most popularly used formulations as many valuable properties and statistics can be obtained from them (Chmiela et al., 2017; Stokes et al., 2020). However, as far as we know, most previous works focus on designing neural network models for either 2D or 3D structures, making the model learned in one form fail to be applied in tasks of the other form.

We argue that a general-purpose neural network model in chemistry should at least be able to handle molecular tasks across data modalities. In this paper, we take the first step toward this goal by

---

*These two authors contributed equally to this project
†Correspondence to: Di He <`dihe@pku.edu.cn`> and Liwei Wang <`wanglw@pku.edu.cn`>.

developing Transformer-M, a versatile Transformer-based Molecular model that performs well for both 2D and 3D molecular representation learning. Note that for a molecule, its 2D and 3D forms describe the same collection of atoms but use different characterizations of the structure. Therefore, the key challenge is to design a model expressive and compatible in capturing structural knowledge in different formulations and train the parameters to learn from both information. Transformer is more favorable than other architectures as it can explicitly plug structural signals in the model as bias terms (e.g., positional encodings (Vaswani et al., 2017; Raffel et al., 2020)). We can conveniently set 2D and 3D structural information as different bias terms through separated channels and incorporate them with the atom features in the attention layers.

**Architecture.** The backbone network of our Transformer-M is composed of standard Transformer blocks. We develop two separate channels to encode 2D and 3D structural information. The 2D channel uses degree encoding, shortest path distance encoding, and edge encoding extracted from the 2D graph structure, following Ying et al. (2021a). The shortest path distance encoding and edge encoding reflect the spatial relations and bond features of a pair of atoms and are used as bias terms in the softmax attention. The degree encoding is added to the atom features in the input layer. For the 3D channel, we follow Shi et al. (2022) to use the 3D distance encoding to encode the spatial distance between atoms in the 3D geometric structure. Each atom pair's Euclidean distance is encoded via the Gaussian Basis Kernel function (Scholkopf et al., 1997) and will be used as a bias term in the softmax attention. For each atom, we sum up the 3D distance encodings between it and all other atoms, and add it to atom features in the input layer. See Figure 1 for an illustration.

**Training.** Except for the parameters in the two structural channels, all other parameters in Transformer-M (e.g., self-attention and feed-forward networks) are shared for different data modalities. We design a joint-training approach for Transformer-M to learn its parameters. During training, when the instances in a batch are only associated with 2D graph structures, the 2D channel will be activated, and the 3D channel will be disabled. Similarly, when the instances in a batch use 3D geometric structures, the 3D channel will be activated, and the 2D channel will be disabled. When both 2D and 3D information are given, both channels will be activated. In such a way, we can collect 2D and 3D data from separate databases and train Transformer-M with different training objectives, making the training process more flexible. We expect a single model to learn to identify and incorporate information from different modalities and efficiently utilize the parameters, leading to better generalization performance.

**Experimental Results.** We use the PCQM4Mv2 dataset in the OGB Large-Scale Challenge (OGB-LSC) (Hu et al., 2021) to train our Transformer-M, which consists of 3.4 million molecules of both 2D and 3D forms. The model is trained to predict the pre-computed HOMO-LUMO gap of each data instance in different formats with a pre-text 3D denoising task specifically for 3D data. With the pre-trained model, we directly use or fine-tune the parameters for various molecular tasks of different data formats. First, we show that on the validation set of the PCQM4Mv2 task, which only contains 2D molecular graphs, our Transformer-M surpasses all previous works by a large margin. The improvement is credited to the joint training, which effectively mitigates the overfitting problem. Second, On PDBBind (Wang et al., 2004; 2005b) (2D&3D), the fine-tuned Transformer-M achieves state-of-the-art performance compared to strong baselines. Lastly, on QM9 (Ramakrishnan et al., 2014) (3D) benchmark, the fine-tuned Transformer-M models achieve competitive performance compared to recent methods. All results show that our Transformer-M has the potential to be used as a general-purpose model in a broad range of applications in chemistry.

## 2 RELATED WORKS

**Neural networks for learning 2D molecular representations.** Graph Neural Network (GNN) is popularly used in molecular graph representation learning (Kipf & Welling, 2016; Hamilton et al., 2017; Gilmer et al., 2017; Xu et al., 2019; Veličković et al., 2018). A GNN learns node and graph representations by recursively aggregating (i.e., message passing) and updating the node representations from neighbor representations. Different architectures are developed by using different aggregation and update strategies. We refer the readers to Wu et al. (2020) for a comprehensive survey. Recently, many works extended the Transformer model to graph tasks (Dwivedi & Bresson, 2020; Kreuzer et al., 2021; Ying et al., 2021a; Luo et al., 2022; Kim et al., 2022; Rampášek et al., 2022; Park et al., 2022; Hussain et al., 2022; Zhang et al., 2023). Seminal works include Graphormer (Ying et al., 2021a), which developed graph structural encodings and used them in a standard Transformer model.

**Neural networks for learning 3D molecular representations.** Learning molecular representations with 3D geometric information is essential in many applications, such as molecular dynamics simulation. Recently, researchers have designed architectures to preserve invariant and equivariant properties for several necessary transformations like rotation and translation. Schütt et al. (2017) used continuous-filter convolutional layers to model quantum interactions in molecules. Thomas et al. (2018) used filters built from spherical harmonics to construct a rotation- and translation-equivariant neural network. Klicpera et al. (2020) proposed directional message passing, which ensured their embeddings to be rotationally equivariant. Liu et al. (2022); Wang et al. (2022) use spherical coordinates to capture geometric information and achieve equivariance. Hutchinson et al. (2021); Thölke & De Fabritiis (2021) built Transformer models preserving equivariant properties. Shi et al. (2022) extended Ying et al. (2021a) to a 3D Transformer model which attains better results on large-scale molecular modeling challenges (Chanussot et al., 2020).

**Multi-view learning for molecules.** The 2D graph structure and 3D geometric structure can be considered as different views of the same molecule. Inspired by the contrastive pre-training approach in vision (Chen et al., 2020; He et al., 2020; Radford et al., 2021), many works studied pre-training methods for molecules by jointly using the 2D and 3D information. Stärk et al. (2022) used two encoders to encode the 2D and 3D molecular information separately while maximizing the mutual information between the representations. Liu et al. (2021a) derived the GraphMVP framework, which uses contrastive learning and reconstruction to pre-train a 2D encoder and a 3D encoder. Zhu et al. (2022) unified the 2D and 3D pre-training methods above and proposed a 2D GNN model that can be enhanced by 3D geometric features. Different from these works, we aim to develop a single model which is compatible with both 2D and 3D molecular tasks. Furthermore, all the above works train models using paired 2D and 3D data, while such paired data is not a strong requirement to train our model.

**General-purpose models.** Building a single agent that works for multiple tasks, even across modalities, is a recent discovery in deep learning. In the early years, researchers found that a single multilingual translation model can translate tens of languages using the same weights and perform better than a bilingual translation model for rare languages (Lample & Conneau, 2019; Conneau et al., 2019; Xue et al., 2020; Liu et al., 2020). Large-scale language model (Devlin et al., 2019; Brown et al., 2020) is another example that can be applied to different downstream tasks using in-context learning or fine-tuning. Reed et al. (2022) further pushed the boundary by building a single generalist agent, Gato. This agent uses the same network with the same weights but can play Atari, caption images, and make conversations like a human. Our work also lies in this direction. We focus on developing a general-purpose model in chemistry, which can take molecules in different formats as input and perform well on various molecular tasks with a small number of additional training data.

## 3 TRANSFORMER-M

In this section, we introduce Transformer-M, a versatile Transformer serving as a general architecture for 2D and 3D molecular representation learning. First, we introduce notations and recap the preliminaries in the backbone Transformer architecture (Section 3.1). After that, we present the proposed Transformer-M model with two structural channels for different data modalities (Section 3.2).

### 3.1 NOTATIONS AND THE BACKBONE TRANSFORMER

A molecule $\mathcal{M}$ is made up of a collection of atoms held together by attractive forces. We denote $\boldsymbol{X} \in \mathbb{R}^{n \times d}$ as the atoms with features, where $n$ is the number of atoms, and $d$ is the feature dimension. The structure of $\mathcal{M}$ can be represented in different formulations, such as 2D graph structure and 3D geometric structure. For the 2D graph structure, atoms are explicitly connected by chemical bonds, and we define $\mathcal{M}^{2D} = (\boldsymbol{X}, E)$, where $\mathbf{e}_{(i,j)} \in E$ denotes the edge feature (i.e., the type of the bond) between atom $i$ and $j$ if the edge exists. For the 3D geometric structure, for each atom $i$, its position $\mathbf{r}_i$ in the Cartesian coordinate system is provided. We define $\mathcal{M}^{3D} = (\boldsymbol{X}, R)$, where $R = \{\mathbf{r}_1, ..., \mathbf{r}_n\}$ and $\mathbf{r}_i \in \mathbb{R}^3$. Our goal is to design a parametric model which can take either $\mathcal{M}^{2D}$ or $\mathcal{M}^{3D}$ (or both of them) as input, obtain contextual representations, and make predictions on downstream tasks.

**Transformer layer.** The backbone architecture we use in this work is the Transformer model (Vaswani et al., 2017). A Transformer is composed of stacked Transformer blocks. A Transformer block consists of two layers: a self-attention layer followed by a feed-forward layer, with both layers having normalization (e.g., LayerNorm (Ba et al., 2016)) and skip connections (He et al., 2016). Denote $\boldsymbol{X}^{(l)}$ as the input to the $(l+1)$-th block and define $\boldsymbol{X}^{(0)} = \boldsymbol{X}$. For an input $\boldsymbol{X}^{(l)}$, the $(l+1)$-th block works as follows:

$$\boldsymbol{A}^h(\boldsymbol{X}^{(l)}) = \quad \text{softmax}\left(\frac{\boldsymbol{X}^{(l)}\boldsymbol{W}_Q^{l,h}(\boldsymbol{X}^{(l)}\boldsymbol{W}_K^{l,h})^\top}{\sqrt{d}}\right); \tag{1}$$

$$\hat{\boldsymbol{X}}^{(l)} = \quad \boldsymbol{X}^{(l)} + \sum_{h=1}^{H} \boldsymbol{A}^h(\boldsymbol{X}^{(l)})\boldsymbol{X}^{(l)}\boldsymbol{W}_V^{l,h}\boldsymbol{W}_O^{l,h}; \tag{2}$$

$$\boldsymbol{X}^{(l+1)} = \quad \hat{\boldsymbol{X}}^{(l)} + \text{GELU}(\hat{\boldsymbol{X}}^{(l)}\boldsymbol{W}_1^l)\boldsymbol{W}_2^l, \tag{3}$$

where $\boldsymbol{W}_O^{l,h} \in \mathbb{R}^{d_H \times d}$, $\boldsymbol{W}_Q^{l,h}, \boldsymbol{W}_K^{l,h}, \boldsymbol{W}_V^{l,h} \in \mathbb{R}^{d \times d_H}$, $\boldsymbol{W}_1^l \in \mathbb{R}^{d \times r}$, $\boldsymbol{W}_2^l \in \mathbb{R}^{r \times d}$. $H$ is the number of attention heads, $d_H$ is the dimension of each head, and $r$ is the dimension of the hidden layer. $\boldsymbol{A}^h(\boldsymbol{X})$ is usually referred to as the attention matrix.

**Positional encoding.** Another essential component in the Transformer is positional encoding. Note that the self-attention layer and the feed-forward layer do not make use of the order of input elements (e.g., word tokens), making the model impossible to capture the structural information. The original paper (Vaswani et al., 2017) developed effective positional encodings to encode the sentence structural information and explicitly integrate them as bias terms into the model. Shortly, many works realized that positional encoding plays a crucial role in extending standard Transformer to more complicated data structures beyond language. By carefully designing structural encoding using domain knowledge, Transformer has successfully been applied to the image and graph domain and achieved impressive performance (Dosovitskiy et al., 2020; Liu et al., 2021b; Ying et al., 2021a).

## 3.2 TRANSFORMER-M AND TRAINING STRATEGY

As we can see, the two molecular formulations defined in Section 3.1 use the same atom feature space but different characterizations of the structure (graph structure $E$ v.s. geometric structure $R$). Therefore, the key challenge is to design a compatible architecture that can utilize either structural information in $E$ or $R$ (or both) and incorporate them with the atom features in a principled way.

The Transformer is a suitable backbone to achieve the goal as we can encode structural information as bias terms and properly plug them into different modules. Furthermore, with Transformer, we can treat $E$ and $R$ in a unified way by decomposing the structural information into pair-wise and atom-wise encodings. Without loss of generality, we choose to use the encoding strategies in the graph and geometric Transformers proposed by Ying et al. (2021a); Shi et al. (2022). For the sake of completeness, we briefly introduce those structural encodings and show how to leverage them in Transformer-M. Note that our design methodology also works with other encoding strategies (Hussain et al., 2022; Park et al., 2022; Thölke & De Fabritiis, 2021). See Appendix B.5 for the detailed results.

**Encoding pair-wise relations in $E$.** We use two terms to encode the structural relations between any atom pairs in the graph. First, we encode the shortest path distance (SPD) between two atoms to reflect their spatial relation. Let $\Phi_{ij}^{\text{SPD}}$ denote the SPD encoding between atom $i$ and $j$, which is a learnable scalar determined by the distance of the shortest path between $i$ and $j$. Second, we encode the edge features (e.g., the chemical bond types) along the shortest path between $i$ and $j$ to reflect the bond information. For most molecules, there exists only one distinct shortest path between any two atoms. Denote the edges in the shortest path from $i$ to $j$ as $\text{SP}_{ij} = (\mathbf{e}_1, \mathbf{e}_2, ..., \mathbf{e}_N)$, and the edge encoding between $i$ and $j$ is defined as $\Phi_{ij}^{\text{Edge}} = \frac{1}{N}\sum_{n=1}^{N} \mathbf{e}_n(w_n)^T$, where $w_n$ are learnable vectors of the same dimension as the edge feature. Denote $\Phi^{\text{SPD}}$ and $\Phi^{\text{Edge}}$ as the matrix form of the SPD encoding and edge encoding, both of which are of shape $n \times n$.

**Encoding pair-wise relations in $R$.** We encode the Euclidean distance to reflect the spatial relation between any pair of atoms in the 3D space. For each atom pair $(i, j)$, we first process their Euclidean distance with the Gaussian Basis Kernel function (Scholkopf et al., 1997), $\psi_{(i,j)}^k =$
$-\frac{1}{\sqrt{2\pi}|\sigma^k|}\exp\left(-\frac{1}{2}\left(\frac{\gamma_{(i,j)}\|\mathbf{r}_i - \mathbf{r}_j\| + \beta_{(i,j)} - \mu^k}{|\sigma^k|}\right)^2\right), k = 1, ..., K$, where $K$ is the number of Gaussian Basis kernels. Then the 3D Distance encoding $\Phi_{ij}^{\text{3D Distance}}$ is obtained according to $\Phi_{ij}^{\text{3D Distance}} = \text{GELU}\left(\boldsymbol{\psi}_{(i,j)}\boldsymbol{W}_D^1\right)\boldsymbol{W}_D^2$, where $\quad \boldsymbol{\psi}_{(i,j)} = [\psi_{(i,j)}^1; ...; \psi_{(i,j)}^K]^\top$, $\boldsymbol{W}_D^1 \in \mathbb{R}^{K \times K}$, $\boldsymbol{W}_D^2 \in \mathbb{R}^{K \times 1}$ are learnable parameters. $\gamma_{(i,j)}, \beta_{(i,j)}$ are learnable scalars indexed by the pair of atom types, and $\mu^k, \sigma^k$ are learnable kernel center and learnable scaling factor of the $k$-th Gaussian Basis Kernel. Denote $\Phi^{\text{3D Distance}}$ as the matrix form of the 3D distance encoding, whose shape is $n \times n$.

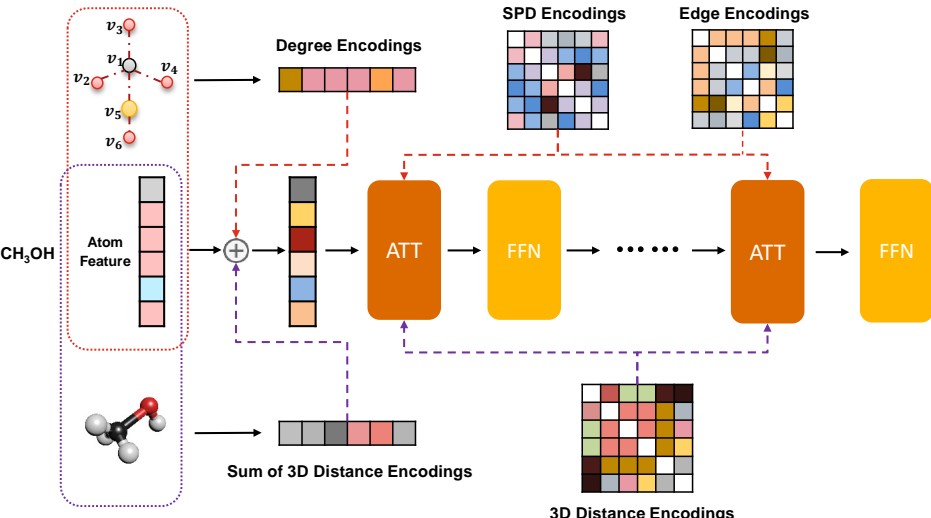

Figure 1: An illustration of our Transformer-M model architecture. We build two channels on the backbone Transformer. The red channel is activated for data with 2D graph structures to incorporate degree, shortest path distance, and edge information. The purple channel is activated for data with 3D geometric structures to leverage Euclidean distance information. Different encodings are located in appropriate modules.

**Integrating $\Phi^{\text{SPD}}$, $\Phi^{\text{Edge}}$ and $\Phi^{\text{3D Distance}}$ in Transformer-M.** All pair-wise encodings defined above capture the interatomic information, which is in a similar spirit to the relative positional encoding for sequential tasks (Raffel et al., 2020). Therefore, we similarly locate those pair-wise signals in the self-attention module to provide complementary information to the dot-product term $\boldsymbol{XW}_Q(\boldsymbol{XW}_K)^\top$. For simplicity, we omit the index of attention head $h$ and layer $l$, and the modified attention matrix is defined as:

$$\boldsymbol{A}(\boldsymbol{X}) = \text{softmax}\left( \frac{\boldsymbol{XW}_Q(\boldsymbol{XW}_K)^\top}{\sqrt{d}} + \underbrace{\Phi^{\text{SPD}} + \Phi^{\text{Edge}}}_{\text{2D pair-wise channel}} + \underbrace{\Phi^{\text{3D Distance}}}_{\text{3D pair-wise channel}} \right) \quad (4)$$

**Encoding atom-wise structural information in $E$.** For atom $i$, Eqn. (4) computes the normalized weights according to the semantic (first term) and spatial relation (last three terms) between $i$ and other atoms. However, the information is still not sufficient. For example, the importance (i.e., centrality) of each atom is missing in the attention. For each atom $i$, we use its degree as the centrality information. Formally, let $\Psi_i^{\text{Degree}}$ denote the degree encoding of the atom $i$, which is a $d$-dimensional learnable vector determined by the degree of the atom. Denote $\Psi^{\text{Degree}} = [\Psi_1^{\text{Degree}}, \Psi_2^{\text{Degree}}, ..., \Psi_n^{\text{Degree}}]$ as the centrality encoding of all the atoms, which is of shape $n \times d$.

**Encoding atom-wise structural information in $R$.** Similar to the 2D atom-wise centrality encoding, for geometric data, we encode the centrality of each atom in the 3D space. For each atom $i$, we sum up the 3D Distance encodings between it and all other atoms. Let $\Psi_i^{\text{Sum of 3D Distance}}$ denote the centrality encoding of atom $i$, we have $\Psi_i^{\text{Sum of 3D Distance}} = \sum_{j \in [n]} \boldsymbol{\psi}_{(i,j)} \boldsymbol{W}_D^3$, where $W_D^3 \in \mathbb{R}^{K \times d}$ is a learnable weight matrix. Similarly, we define $\Psi^{\text{Sum of 3D Distance}}$ as the encoding of all atoms, whose shape is $n \times d$.

**Integrating $\Psi^{\text{Degree}}$ and $\Psi^{\text{Sum of 3D Distance}}$ in Transformer-M.** We add the atom-wise encodings of 2D and 3D structures to the atom features in the input layer. Formally, the input $\boldsymbol{X}^{(0)}$ is modified as:

$$\boldsymbol{X}^{(0)} = \boldsymbol{X} + \underbrace{\Psi^{\text{Degree}}}_{\text{2D atom-wise channel}} + \underbrace{\Psi^{\text{Sum of 3D Distance}}}_{\text{3D atom-wise channel}}, \quad (5)$$

Through this simple way, the structural information of molecules in both 2D and 3D formats is integrated into one Transformer model. It is easy to check that Transformer-M preserves equivariant properties for both data formats.

**Training.**   The next step is to learn the parameters in Transformer-M to capture meaningful representations from each data format. To achieve this goal, we develop a simple and flexible joint training method to learn Transformer-M. We first collect datasets in different formats (2D/3D) and define supervised/self-supervised tasks (e.g., energy regression) on each format, and train the model on all the data toward each objective, respectively. To be concrete, during training, if a data instance comes from a dataset in the 2D format, the 2D channel is activated, and the 3D channel is disabled. The model parameter will be optimized to minimize the corresponding (i.e., 2D) objective. When a data instance comes from a dataset in the 3D format, only the 3D channel is activated, and the model will learn to minimize the 3D objective. Both channels are activated if the model takes molecules in both 2D and 3D formats as input. Compared with the multi-view learning approaches, we can train Transformer-M using unpaired 2D and 3D data, making the training process more flexible.

The Transformer-M may generalize better due to the joint training. Several previous works (Liu et al., 2021a) observed that 2D graph structure and 3D geometric structure contain complementary chemical knowledge. For example, the 2D graph structure only contains bonds with bond type, while the 3D geometric structure contains fine-grained information such as lengths and angles. As another example, the 3D geometric structures are usually obtained from computational simulations like Density Functional Theory (DFT) (Burke, 2012), which could have approximation errors. The 2D graphs are constructed by domain experts, which to some extent, provide references to the 3D structure. By jointly training using 2D and 3D data with parameter sharing, our model can learn more chemical knowledge instead of overfitting to data noise and perform better on both 2D and 3D tasks.

**Future Directions.**   As an initial attempt, our Transformer-M opens up a way to develop general-purpose molecular models to handle diverse chemical tasks in different data formats. We believe it is a starting point with more possibilities to explore in the future. For example, in this work, we use a simple way and linearly combine the structural information of 2D and 3D structures, and we believe there should be other efficient ways to fuse such encodings. Our model can also be combined with previous multi-view contrastive learning approaches. It is worth investigating how to pre-train our model using those methods.

## 4   EXPERIMENTS

In this section, we empirically study the performance of Transformer-M. First, we pre-train our model on the PCQM4Mv2 training set from OGB Large-Scale Challenge (Hu et al., 2021) (Section 4.1). With the pre-trained model, we conduct experiments on molecular tasks in different data formats and evaluate the versatility and effectiveness of our Transformer-M. Due to space limitation, we study three representative tasks, PCQM4Mv2 (2D, Section 4.2), PDBBind (2D & 3D, Section 4.3) and QM9 (3D, Section 4.4). Ablation studies are presented in Section 4.5. All codes are implemented based on the official codebase of Graphormer (Ying et al., 2021a) in PyTorch (Paszke et al., 2019).

### 4.1   LARGE-SCALE PRE-TRAINING

Our Transformer-M is pre-trained on the training set of PCQM4Mv2 from the OGB Large-Scale Challenge (Hu et al., 2021). The total number of training samples is 3.37 million. Each molecule is associated with its 2D graph structure and 3D geometric structure. The HOMO-LUMO energy gap of each molecule is provided as its label, which is obtained by DFT-based methods (Burke, 2012).

We follow Ying et al. (2021a) and employ a 12-layer Transformer-M model. The dimension of hidden layers and feed-forward layers is set to 768. The number of attention heads is set to 32. The number of Gaussian Basis kernels is set to 128. To train Transformer-M, we provide three modes for each data instance: (1) activate the 2D channels and disable the 3D channels (2D mode); (2) activate the 3D channels and disable the 2D channels (3D mode); (3) activate both channels (2D+3D mode). The mode of each data instance during training is randomly drawn on the fly according to a pre-defined distribution, implemented similarly to Dropout (Srivastava et al., 2014). In this work, we use two training objectives. The first one is a supervised learning objective, which aims to predict the HOMO-LUMO energy gap of each molecule. Besides, we also use a self-supervised learning objective called 3D Position Denoising (Godwin et al., 2022; Zaidi et al., 2022), which is particularly effective. During training, if a data instance is in the 3D mode, we add Gaussian noise to the position of each atom and require the model to predict the noise from the noisy input. The model is optimized to minimize a linear combination of the two objectives above. Details of settings are in Appendix B.1.

Table 1: Results on PCQM4Mv2 validation set in OGB Large-Scale Challenge (Hu et al., 2021). The evaluation metric is the Mean Absolute Error (MAE) [eV]. We report the official results of baselines from OGB and use ∗ to indicate our implemented results. Bold values indicate the best performance.

| method | #param. | Valid MAE |
|---|---|---|
| MLP-Fingerprint (Hu et al., 2021) | 16.1M | 0.1753 |
| GCN (Kipf & Welling, 2016) | 2.0M | 0.1379 |
| GIN (Xu et al., 2019) | 3.8M | 0.1195 |
| GINE-VN (Brossard et al., 2020; Gilmer et al., 2017) | 13.2M | 0.1167* |
| GCN-VN (Kipf & Welling, 2016; Gilmer et al., 2017) | 4.9M | 0.1153 |
| GIN-VN (Xu et al., 2019; Gilmer et al., 2017) | 6.7M | 0.1083 |
| DeeperGCN-VN (Li et al., 2020; Gilmer et al., 2017) | 25.5M | 0.1021* |
| GraphGPS$_{SMALL}$ (Rampášek et al., 2022) | 6.2M | 0.0938 |
| CoAtGIN (Cui, 2022) | 5.2M | 0.0933 |
| TokenGT (Kim et al., 2022) | 48.5M | 0.0910 |
| GRPE$_{BASE}$ (Park et al., 2022) | 46.2M | 0.0890 |
| EGT (Hussain et al., 2022) | 89.3M | 0.0869 |
| GRPE$_{LARGE}$ (Park et al., 2022) | 46.2M | 0.0867 |
| Graphormer (Ying et al., 2021a; Shi et al., 2022) | 47.1M | 0.0864 |
| GraphGPS$_{BASE}$ (Rampášek et al., 2022) | 19.4M | 0.0858 |
| Transformer-M (ours) | 47.1M | **0.0787** |

## 4.2 PCQM4Mv2 Performance (2D)

After the model is pre-trained, we evaluate our Transformer-M on the validation set of PCQM4Mv2. Note that the validation set of PCQM4Mv2 consists of molecules in the 2D format only. Therefore, we can use it to evaluate how well Transformer-M performs on 2D molecular data. The goal of the task is to predict the HOMU-LUMO energy gap, and the evaluation metric is the Mean Absolute Error (MAE). As our training objectives include the HOMO-LUMO gap prediction task, we didn't fine-tune the model parameters on any data. During inference, only the 2D channels are activated. We choose several strong baselines covering message passing neural network (MPNN) variants and Graph Transformers. Detailed descriptions of baselines are presented in Appendix B.2.

The results are shown in Table 1. It can be easily seen that our Transformer-M surpasses all baselines by a large margin, e.g., $8.2\%$ relative MAE reduction compared to the previous best model (Rampášek et al., 2022), establishing a new state-of-the-art on PCQM4Mv2 dataset. Note that our general architecture is the same as the Graphormer model (Ying et al., 2021a). The only difference between Transformer-M and the Graphormer baseline is that Graphormer is trained on 2D data only, while Transformer-M is trained using both 2D and 3D structural information. Therefore, we can conclude that Transformer-M performs well on 2D molecular data, and the 2D-3D joint training with shared parameters indeed helps the model learn more chemical knowledge.

## 4.3 PDBBind Performance (2D & 3D)

To verify the compatibility of our Transformer-M, we further fine-tune our model on the PDBBind dataset (version 2016, Wang et al. (2004; 2005b)), one of the most widely used datasets for structure-based virtual screening (Jiménez et al., 2018; Stepniewska-Dziubinska et al., 2018; Zheng et al., 2019). PDBBind dataset consists of protein-ligand complexes as data instances, which are obtained in bioassay experiments associated with the $pK_a$ (or $-\log K_d$, $-\log K_i$) affinity values. For each data instance, the 3D geometric structures are provided and the 2D graph structures are constructed via pre-defined rules. The task requires models to predict the binding affinity of protein-ligand complexes, which is extremely vital for drug discovery. After pre-trained on the PCQM4Mv2 training set, our Transformer-M model is fine-tuned and evaluated on the core set of the PDBBind dataset.

We compare our model with competitive baselines covering classical methods, CNN-based methods, and GNNs. All experiments are repeated five times with different seeds. Average performance is reported. Due to space limitation, we present the details of baselines and experiment settings in Appendix B.3. The results are presented in Table 2. Our Transformer-M consistently outperforms all the baselines on all evaluation metrics by a large margin, e.g., $3.3\%$ absolute improvement on Pearson's correlation coefficient (R). It is worth noting that data instances of the PDBBind dataset are protein-ligand complexes, while our model is pre-trained on simple molecules, demonstrating the transferability of Transformer-M.

Table 2: Results on PDBBind core set (version 2016) (Wang et al., 2004; 2005b). The evaluation metrics include Pearson's correlation coefficient (R), Mean Absolute Error (MAE), Root-Mean Squared Error (RMSE), and Standard Deviation (SD). We report the official results of baselines from Li et al. (2021). Bold values indicate the best performance.

| Method | PDBBind core set | | | |
| --- | --- | --- | --- | --- |
| | R $\uparrow$ | MAE $\downarrow$ | RMSE $\downarrow$ | SD $\downarrow$ |
| LR | $0.671_{\pm 0.000}$ | $1.358_{\pm 0.000}$ | $1.675_{\pm 0.000}$ | $1.612_{\pm 0.000}$ |
| SVR | $0.727_{\pm 0.000}$ | $1.264_{\pm 0.000}$ | $1.555_{\pm 0.000}$ | $1.493_{\pm 0.000}$ |
| RF-Score (Ballester et al., 2010) | $0.789_{\pm 0.003}$ | $1.161_{\pm 0.007}$ | $1.446_{\pm 0.008}$ | $1.335_{\pm 0.010}$ |
| Pafnucy (Stepniewska-Dziubinska et al., 2018) | $0.695_{\pm 0.011}$ | $1.284_{\pm 0.021}$ | $1.585_{\pm 0.013}$ | $1.563_{\pm 0.022}$ |
| OnionNet (Zheng et al., 2019) | $0.768_{\pm 0.014}$ | $1.078_{\pm 0.028}$ | $1.407_{\pm 0.034}$ | $1.391_{\pm 0.038}$ |
| GraphDTA$_{GCN}$ (Nguyen et al., 2020) | $0.613_{\pm 0.016}$ | $1.343_{\pm 0.037}$ | $1.735_{\pm 0.034}$ | $1.719_{\pm 0.027}$ |
| GraphDTA$_{GAT}$ (Nguyen et al., 2020) | $0.601_{\pm 0.016}$ | $1.354_{\pm 0.033}$ | $1.765_{\pm 0.026}$ | $1.740_{\pm 0.027}$ |
| GraphDTA$_{GIN}$ (Nguyen et al., 2020) | $0.667_{\pm 0.018}$ | $1.261_{\pm 0.044}$ | $1.640_{\pm 0.044}$ | $1.621_{\pm 0.036}$ |
| GraphDTA$_{GAT-GCN}$ (Nguyen et al., 2020) | $0.697_{\pm 0.008}$ | $1.191_{\pm 0.016}$ | $1.562_{\pm 0.022}$ | $1.558_{\pm 0.018}$ |
| GNN-DTI (Lim et al., 2019) | $0.736_{\pm 0.021}$ | $1.192_{\pm 0.032}$ | $1.492_{\pm 0.025}$ | $1.471_{\pm 0.051}$ |
| DMPNN (Yang et al., 2019) | $0.729_{\pm 0.006}$ | $1.188_{\pm 0.009}$ | $1.493_{\pm 0.016}$ | $1.489_{\pm 0.014}$ |
| SGCN (Danel et al., 2020) | $0.686_{\pm 0.015}$ | $1.250_{\pm 0.036}$ | $1.583_{\pm 0.033}$ | $1.582_{\pm 0.320}$ |
| MAT (Maziarka et al., 2020) | $0.747_{\pm 0.013}$ | $1.154_{\pm 0.037}$ | $1.457_{\pm 0.037}$ | $1.445_{\pm 0.033}$ |
| DimeNet (Klicpera et al., 2020) | $0.752_{\pm 0.010}$ | $1.138_{\pm 0.026}$ | $1.453_{\pm 0.027}$ | $1.434_{\pm 0.023}$ |
| CMPNN (Song et al., 2020) | $0.765_{\pm 0.009}$ | $1.117_{\pm 0.031}$ | $1.408_{\pm 0.028}$ | $1.399_{\pm 0.025}$ |
| SIGN (Li et al., 2021) | $0.797_{\pm 0.012}$ | $1.027_{\pm 0.025}$ | $1.316_{\pm 0.031}$ | $1.312_{\pm 0.035}$ |
| Transformer-M (ours) | $\mathbf{0.830}_{\pm 0.011}$ | $\mathbf{0.940}_{\pm 0.006}$ | $\mathbf{1.232}_{\pm 0.013}$ | $\mathbf{1.207}_{\pm 0.007}$ |

Table 3: Results on QM9 (Ramakrishnan et al., 2014). The evaluation metric is the Mean Absolute Error (MAE). We report the official results of baselines from Thölke & De Fabritiis (2021); Godwin et al. (2022); Jiao et al. (2022). Bold values indicate the best performance.

| method | $\mu$ | $\alpha$ | $\epsilon_{HOMO}$ | $\epsilon_{LUMO}$ | $\Delta\epsilon$ | $R^2$ | ZPVE | $U_0$ | U | H | G | $C_v$ |
| --- | --- | --- | --- | --- | --- | --- | --- | --- | --- | --- | --- | --- |
| EdgePred (Hamilton et al., 2017) | 0.039 | 0.086 | 37.4 | 31.9 | 58.2 | 0.112 | 1.81 | 14.7 | 14.2 | 14.8 | 14.5 | 0.038 |
| AttrMask (Hu et al., 2019) | 0.020 | 0.072 | 31.3 | 37.8 | 50.0 | 0.423 | 1.90 | 10.7 | 10.8 | 11.4 | 11.2 | 0.062 |
| InfoGraph (Sun et al., 2019) | 0.041 | 0.099 | 48.1 | 38.1 | 72.2 | 0.114 | 1.69 | 16.4 | 14.9 | 14.5 | 16.5 | 0.030 |
| GraphCL (You et al., 2020) | 0.027 | 0.066 | 26.8 | 22.9 | 45.5 | 0.095 | 1.42 | 9.6 | 9.7 | 9.6 | 10.2 | 0.028 |
| GPT-GNN (Hu et al., 2020b) | 0.039 | 0.103 | 35.7 | 28.8 | 54.1 | 0.158 | 1.75 | 12.0 | 24.8 | 14.8 | 12.2 | 0.032 |
| GraphMVP (Jing et al., 2021) | 0.031 | 0.070 | 28.5 | 26.3 | 46.9 | 0.082 | 1.63 | 10.2 | 10.3 | 10.4 | 11.2 | 0.033 |
| GEM (Fang et al., 2021) | 0.034 | 0.081 | 33.8 | 27.7 | 52.1 | 0.089 | 1.73 | 13.4 | 12.6 | 13.3 | 13.2 | 0.035 |
| 3D Infomax (Stärk et al., 2022) | 0.034 | 0.075 | 29.8 | 25.7 | 48.8 | 0.122 | 1.67 | 12.7 | 12.5 | 12.4 | 13.0 | 0.033 |
| PosPred (Jiao et al., 2022) | 0.024 | 0.067 | 25.1 | 20.9 | 40.6 | 0.115 | 1.46 | 10.2 | 10.3 | 10.2 | 10.9 | 0.035 |
| 3D-MGP (Jiao et al., 2022) | 0.020 | 0.057 | 21.3 | 18.2 | 37.1 | 0.092 | 1.38 | 8.6 | 8.6 | 8.7 | 9.3 | 0.026 |
| Schnet (Schütt et al., 2017) | 0.033 | 0.235 | 41 | 34 | 63 | 0.073 | 1.7 | 14 | 19 | 14 | 14 | 0.033 |
| PhysNet (Unke & Meuwly, 2019) | 0.0529 | 0.0615 | 32.9 | 24.7 | 42.5 | 0.765 | 1.39 | 8.15 | 8.34 | 8.42 | 9.4 | 0.028 |
| Cormorant (Anderson et al., 2019) | 0.038 | 0.085 | 34 | 38 | 61 | 0.961 | 2.027 | 22 | 21 | 21 | 20 | 0.026 |
| DimeNet++ (Klicpera et al., 2020) | 0.0297 | 0.0435 | 24.6 | 19.5 | 32.6 | 0.331 | 1.21 | 6.32 | 6.28 | 6.53 | 7.56 | 0.023 |
| PaiNN (Schütt et al., 2021) | 0.012 | 0.045 | 27.6 | 20.4 | 45.7 | 0.066 | 1.28 | $\mathbf{5.85}$ | $\mathbf{5.83}$ | $\mathbf{5.98}$ | $\mathbf{7.35}$ | 0.024 |
| LieTF (Hutchinson et al., 2021) | 0.041 | 0.082 | 33 | 27 | 51 | 0.448 | 2.10 | 17 | 16 | 17 | 19 | 0.035 |
| TorchMD-Net (Thölke & De Fabritiis, 2021) | $\mathbf{0.011}$ | 0.059 | 20.3 | 17.5 | 36.1 | $\mathbf{0.033}$ | 1.84 | 6.15 | 6.38 | 6.16 | 7.62 | 0.026 |
| EGNN (Satorras et al., 2021) | 0.029 | 0.071 | 29 | 25 | 48 | 0.106 | 1.55 | 11 | 12 | 12 | 12 | 0.031 |
| NoisyNode (Godwin et al., 2022) | 0.025 | 0.052 | 20.4 | 18.6 | 28.6 | 0.70 | $\mathbf{1.16}$ | 7.30 | 7.57 | 7.43 | 8.30 | 0.025 |
| Transformer-M (ours) | 0.037 | $\mathbf{0.041}$ | $\mathbf{17.5}$ | $\mathbf{16.2}$ | $\mathbf{27.4}$ | 0.075 | 1.18 | 9.37 | 9.41 | 9.39 | 9.63 | $\mathbf{0.022}$ |

## 4.4 QM9 PERFORMANCE (3D)

We use the QM9 dataset (Ramakrishnan et al., 2014) to evaluate our Transformer-M on molecular tasks in the 3D data format. QM9 is a quantum chemistry benchmark consisting of 134k stable small organic molecules. These molecules correspond to the subset of all 133,885 species out of the GDB-17 chemical universe of 166 billion organic molecules. Each molecule is associated with 12 targets covering its energetic, electronic, and thermodynamic properties. The 3D geometric structure of the molecule is used as input. Following Thölke & De Fabritiis (2021), we randomly choose 10,000 and 10,831 molecules for validation and test evaluation, respectively. The remaining molecules are used to fine-tune our Transformer-M model. We observed that several previous works used different data splitting ratios or didn't describe the evaluation details. For a fair comparison, we choose baselines that use similar splitting ratios in the original papers. The details of baselines and experiment settings are presented in Appendix B.4.

The results are presented in Table 3. It can be seen that our Transformer-M achieves competitive performance compared to those baselines, suggesting that the model is compatible with 3D molecular data. In particular, Transformer-M performs best on HUMO, LUMO, and HUMO-LUMO predictions. This indicates that the knowledge learned in the pre-training task transfers better to similar tasks. Note that the model doesn't perform quite well on some other tasks. We believe the Transformer-M can be improved in several aspects, including employing a carefully designed output layer (Thölke & De Fabritiis, 2021) or pre-training with more self-supervised training signals.

## 4.5 ABLATION STUDY

In this subsection, we conduct a series of experiments to investigate the key designs of our Transformer-M. In this paper, we use two training objectives to train the model, and we will ablate the effect of the two training objectives. Besides, we use three modes to activate different channels with a pre-defined distribution, and we will study the impact of the distribution on the final performance. Due to space limitation, we present more analysis on our Transformer-M model in Appendix B.5.

Table 4: Impact of pre-training tasks and mode distribution on Transformer-M. All other hyperparameters are kept the same for a fair comparison.

| Tasks | | Mode Distribution $(p_{2D}, p_{3D}, p_{2D\&3D})$ | PCQM4Mv2 | QM9 | | |
|---|---|---|---|---|---|---|
| 2D-3D joint Pre-training | 3D Position Denoising | | Valid MAE | $\epsilon_{HOMO}$ | $\epsilon_{LUMO}$ | $\Delta\epsilon$ |
| ✗ | ✗ | - | 0.0878 | 26.5 | 23.8 | 42.3 |
| ✓ | ✗ | 1:2:1 | 0.0811 | 20.1 | 19.3 | 31.2 |
| ✓ | ✓ | 1:2:1 | 0.0787 | 17.5 | 16.2 | 27.4 |
| ✓ | ✓ | 1:1:1 | 0.0796 | 18.1 | 16.9 | 27.9 |
| ✓ | ✓ | 1:2:2 | 0.0789 | 17.8 | 16.6 | 27.5 |

**Impact of the pre-training tasks.** As stated in Section 4.1, our Transformer-M model is pre-trained on the PCQM4Mv2 training set via two tasks: (1) predicting the HOMO-LUMO gap of molecules in both 2D and 3D formats. (2) 3D position denoising. We conduct ablation studies on both PCQM4Mv2 and QM9 datasets to check whether both objectives benefit downstream tasks. In detail, we conduct two additional experiments. The first experiment is training Transformer-M models from scratch on PCQM4Mv2 using its 2D graph data and QM9 using its 3D geometric data to check the benefit of the overall pre-training method. The second experiment is pre-training Transformer-M without using the 3D denoising task to study the effectiveness of the proposed 2D-3D joint pre-training approach. The results are shown in Table 4. It can be seen that the joint pre-training significantly boosts the performance on both PCQM4Mv2 and QM9 datasets. Besides, the 3D Position Denoising task is also beneficial, especially on the QM9 dataset in the 3D format.

**Impact of mode distribution.** Denote $(p_{2D}, p_{3D}, p_{2D\&3D})$ as the probability of the modes mentioned in Section 4.1. We conduct experiments to investigate the influence of different distributions on the model performance. We select three distribution with $(p_{2D}, p_{3D}, p_{2D\&3D})$ being: 1:1:1, 1:2:2, and 1:2:1. The results are presented in Table 4. We obtain consistent conclusions on both PCQM4Mv2 and QM9 datasets: 1) for all three configurations, our Transformer-M model achieves strong performance, which shows that our joint training is robust to hyperparameter selection; 2) Using a slightly larger probability on the 3D mode achieves the best results.

## 5 CONCLUSION

In this work, we take the first step toward general-purpose molecular models. The proposed Transformer-M offers a promising way to handle molecular tasks in 2D and 3D formats. We use two separate channels to encode 2D and 3D structural information and integrate them into the backbone Transformer. When the input data is in a particular format, the corresponding channel will be activated, and the other will be disabled. Through simple training tasks on 2D and 3D molecular data, our model automatically learns to leverage chemical knowledge from different data formats and correctly capture the representations. Extensive experiments are conducted, and all empirical results show that our Transformer-M can achieve strong performance on 2D and 3D tasks simultaneously. The potential of our Transformer-M can be further explored in a broad range of applications in chemistry.

ACKNOWLEDGEMENTS

We thank Shanda Li for the helpful discussions. We also thank all the anonymous reviewers for the very careful and detailed reviews as well as the valuable suggestions. Their help has further enhanced our work. This work is supported by National Key R&D Program of China (2022ZD0114900) and National Science Foundation of China (NSFC62276005). This work is partially supported by the Shanghai Committee of Science and Technology (Grant No. 21DZ1100100).

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

## A  IMPLEMENTATION DETAILS OF TRANSFORMER-M

**2D-3D joint Pre-training.** To effectively utilize molecular data in both 2D and 3D formats, we use a simple strategy. During training, each data instance in a batch has three modes: (1) activate the 2D channels, and disable the 3D channels (2D mode); (2) activate the 3D channels, and disable the 2D channels (3D mode); (3) activate both the 2D and 3D channels (2D+3D mode). The mode of each data instance during training is randomly drawn on the fly according to a pre-defined distribution $(p_{\text{2D}}, p_{\text{3D}}, p_{\text{2D\&3D}})$, implemented similarly to Dropout (Srivastava et al., 2014). For each data instance, the model is required to predict its HOMO-LUMO energy gap across the above three modes.

**Prediction head for position output. (3D Position Denoising)** Following Godwin et al. (2022); Zaidi et al. (2022), we further adopt the 3D Position Denoising task as a self-supervised learning objective. During training, if a data instance is in the 3D mode, we add Gaussian noise to the positions of each atom. The model is required to predict the noise from the noisy input. Formally, let $R = \{\mathbf{r}_1, \mathbf{r}_2, ..., \mathbf{r}_n\}, \mathbf{r}_i \in \mathbb{R}^3$ denote the atom positions of a molecule, and the noisy version of the atom positions is denoted by $\hat{R} = \{\mathbf{r}_1 + \sigma\boldsymbol{\epsilon}_1, \mathbf{r}_2 + \sigma\boldsymbol{\epsilon}_2, ..., \mathbf{r}_n + \sigma\boldsymbol{\epsilon}_n\}$, where $\sigma$ is the scaling factor of noise and $\boldsymbol{\epsilon}_i \sim \mathcal{N}(\mathbf{0}, \mathbf{I})$. The prediction of the model is denoted by $\{\hat{\boldsymbol{\epsilon}}_1, ..., \hat{\boldsymbol{\epsilon}}_n\}$. Following Shi et al. (2022), we use an SE(3) equivariant attention layer as the prediction head:

$$\hat{\boldsymbol{\epsilon}}_i^k = \left(\sum_{v_j \in V} a_{ij}\Delta_{ij}^k \boldsymbol{X}_j^{(L)} \boldsymbol{W}_N^1\right) \boldsymbol{W}_N^2, \quad k = 0, 1, 2 \tag{6}$$

where $\boldsymbol{X}_j^{(L)}$ is the output of the last Transformer block, $a_{ij}$ is the attention score between atom $i$ and $j$ calculated by Eqn.4, $\Delta_{ij}^k$ is the k-th element of the directional vector $\frac{\mathbf{r}_i - \mathbf{r}_j}{\|\mathbf{r}_i - \mathbf{r}_j\|}$ between atom $i$ and $j$, and $\boldsymbol{W}_N^1 \in \mathbb{R}^{d \times d}, \boldsymbol{W}_N^2 \in \mathbb{R}^{d \times 1}$ are learnable weight matrices. The denoising loss of a batch of molecules $\mathcal{E} = \{V^1, ..., V^{|\mathcal{E}|}\}$ is calculated by the cosine similarity between the predicted noises and the ground-truth noises as:

$$\mathcal{L}_{pos} = \frac{1}{|\mathcal{E}|} \sum_{V^i \in \mathcal{E}} \sum_{j \in V^i} \left(1 - \cos\left(\boldsymbol{\epsilon}_j^i, \hat{\boldsymbol{\epsilon}}_j^i\right)\right) \tag{7}$$

**Prediction head for scalar output.** We follow Ying et al. (2021a) to add a pseudo atom to each molecule and make connections between the pseudo atom and other atoms individually. The bias terms in the self-attention layer between this pseudo atom and other atoms are set to the same learnable scalar. For molecule-level prediction, we can simply use the representation of this pseudo atom to predict targets.

## B  EXPERIMENTAL DETAILS

### B.1  LARGE-SCALE PRE-TRAINING

**Dataset.** Our Transformer-M model is pre-trained on the training set of PCQM4Mv2 from the OGB Large-Scale Challenge (Hu et al., 2021). PCQM4Mv2 is a quantum chemistry dataset originally curated under the PubChemQC project (Maho, 2015; Nakata & Shimazaki, 2017). The total number of training samples is 3.37 million. Each molecule in the training set is associated with both 2D graph structures and 3D geometric structures. The HOMO-LUMO energy gap of each molecule is provided, which is obtained by DFT-based geometry optimization (Burke, 2012). According to the OGB-LSC (Hu et al., 2021), the HOMO-LUMO energy gap is one of the most practically-relevant quantum chemical properties of molecules since it is related to reactivity, photoexcitation, and charge transport. Being the largest publicly available dataset for molecular property prediction, PCQM4Mv2 is considered to be a challenging benchmark for molecular models.

**Settings.** Our Transformer-M model consists of 12 layers. The dimension of hidden layers and feed-forward layers is set to 768. The number of attention heads is set to 32. The number of Gaussian Basis kernels is set to 128. We use AdamW (Kingma & Ba, 2014) as the optimizer and set its hyperparameter $\epsilon$ to 1e-8 and $(\beta_1, \beta_2)$ to (0.9,0.999). The gradient clip norm is set to 5.0. The peak learning rate is set to 2e-4. The batch size is set to 1024. The model is trained for 1.5 million steps

with a 90k-step warm-up stage. After the warm-up stage, the learning rate decays linearly to zero. The dropout ratios for the input embeddings, attention matrices, and hidden representations are set to 0.0, 0.1, and 0.0 respectively. The weight decay is set to 0.0. We also employ the stochastic depth (Huang et al., 2016) and set the probability to 0.2. The probability $(p_{2D}, p_{3D}, p_{2D\&3D})$ of each data instance entering the three modes mentioned in Section 4.1 is set to $(0.2, 0.5, 0.3)$. The scaling factor $\sigma$ of added noise in the 3D Position Denoising task is set to 0.2. The ratio of the supervised loss to the denoising loss is set to 1:1. All models are trained on 4 NVIDIA Tesla A100 GPUs.

## B.2 PCQM4Mv2

**Baselines.** We compare our Transformer-M with several competitive baselines. These models fall into two categories: message passing neural network (MPNN) variants and Graph Transformers.

For MPNN variants, we include two widely used models, GCN (Kipf & Welling, 2016) and GIN (Xu et al., 2019), and their variants with virtual node (VN) (Gilmer et al., 2017; Hu et al., 2020a). Additionally, we compare GINE-VN (Brossard et al., 2020) and DeeperGCN-VN (Li et al., 2020). GINE is the multi-hop version of GIN. DeeperGCN is a 12-layer GNN model with carefully designed aggregators. The result of MLP-Fingerprint (Hu et al., 2021) is also reported.

We also compare several Graph Transformer models. Graphormer (Ying et al., 2021a) developed graph structural encodings and integrated them into a standard Transformer model. It achieved impressive performance across several world competitions (Ying et al., 2021b; Shi et al., 2022). CoAt-GIN (Cui, 2022) is a hybrid architecture combining both Convolution and Attention. TokenGT (Kim et al., 2022) adopted the standard Transformer architecture without graph-specific modifications. GraphGPS (Rampášek et al., 2022) proposed a framework to integrate the positional and structural encodings, local message-passing mechanism, and global attention mechanism into the Transformer model. GRPE (Park et al., 2022) proposed a graph-specific relative positional encoding and considerd both node-spatial and node-edge relations. EGT (Hussain et al., 2022) exclusively used global self-attention as an aggregation mechanism rather than static localized convolutional aggregation, and utilized edge channels to capture structural information.

## B.3 PDBBIND

**Dataset.** PDBBind is a well-known dataset that provides a comprehensive collection of experimentally measured binding affinity data for biomolecular complexes deposited in the Protein Data Bank (PDB) (Wang et al., 2005a). The task requires models to predict the binding affinity value $pK_a$ (or $-\log K_d$, $-\log K_i$) of protein-ligand complexes, which is extremely vital for drug discovery. In our experiment, we use the PDBBind v2016 dataset, which is widely used in recent works (Li et al., 2021). The PDBBind dataset includes three overlapped subsets called the general, refined, and core sets. The general set contains all 13,283 protein-ligand complexes, while the 4,057 complexes in the refined set are selected out of the general set with better quality. Moreover, the core set serves as the highest quality benchmark for testing. To avoid data leakage, we remove the data instances in the core set from the refined set. After training, we evaluate our model on the core set. The evaluation metrics include Pearson's correlation coefficient (R), Mean Absolute Error (MAE), Root-Mean Squared Error (RMSE), and Standard Deviation (SD).

**Baselines.** We compare our Transformer-M with several competitive baselines. These models mainly fall into three categories: classic Machine Learning methods, Convolution Neural Network (CNN) based methods, Graph Neural Network (GNN) based methods.

First, we report the results of LR, SVR, and RF-Score (Ballester et al., 2010), which employed traditional machine learning approaches to predict the binding affinities. Second, inspired by the success of CNNs in computer vision, Stepniewska-Dziubinska et al. (2018) proposed the Pafnucy model that represents the complexes via a 3D grid and utilizes 3D convolution to produce feature maps. Zheng et al. (2019) introduced OnionNet, which also used CNNs to extract features based on rotation-free element-pair specific contacts between atoms of proteins and ligands.

There are also several works that leverage GNNs to improve the performance of the PDBBind dataset. GraphDTA (Nguyen et al., 2020) represented protein-ligand complexes as 2D graphs and used GNN models to predict the affinity score. GNN-DTI (Lim et al., 2019) incorporated the 3D structures of protein-ligand complexes into GNNs. DMPNN (Yang et al., 2019) operated over a hybrid representation that combines convolutions and descriptors. SGCN (Danel et al., 2020) is a GCN-inspired architecture that leverages node positions. MAT (Maziarka et al., 2020) augmented the

attention mechanism in the standard Transformer model with inter-atomic distances and molecular graph structures. DimeNet (Klicpera et al., 2020) developed the atom-pair embeddings and utilized directional information between atoms. CMPNN (Song et al., 2020) introduced a communicative kernel and a message booster module to strengthen the message passing between atoms. SIGN (Li et al., 2021)) proposed polar-inspired graph attention layers and pairwise interactive pooling layers to utilize the biomolecular structural information.

**Settings.** We fine-tune the pre-trained Transformer-M on the PDBBind dataset. We use AdamW (Kingma & Ba, 2014) as the optimizer and set its hyperparameter $\epsilon$ to 1e-8 and $(\beta_1, \beta_2)$ to (0.9,0.999). The gradient clip norm is set to 5.0. The peak learning rate is set to 2e-4. The total number of epochs is set to 30. The ratio of the warm-up steps to the total steps is set to 0.06. The batch size is set to 16. The dropout ratios for the input embeddings, attention matrices, and hidden representations are set to 0.0, 0.1, and 0.0 respectively. The weight decay is set to 0.0. Following (Ying et al., 2021a), We use FLAG (Kong et al., 2020) with minor modifications for graph data augmentation. In particular, except for the step size $\alpha$ and the number of adversarial attack steps $m$, we also employ a projection step in Zhu et al. (2020) with maximum perturbation $\gamma$. These hyperparaters are set to the following configurations: $\alpha = 0.01, m = 4, \gamma = 0.01$. All models are trained on 2 NVIDIA Tesla V100 GPUs.

### B.4 QM9

**Dataset.** QM9 (Ramakrishnan et al., 2014) is a quantum chemistry benchmark consisting of 134k stable small organic molecules. These molecules correspond to the subset of all 133,885 species out of the GDB-17 chemical universe of 166 billion organic molecules. Each molecule is associated with 12 targets covering its energetic, electronic, and thermodynamic properties. The 3D geometric structure of the molecule is used as input. Following Thölke & De Fabritiis (2021), we randomly choose 10,000 and 10,831 molecules for validation and test evaluation, respectively. The remaining molecules are used to fine-tune our Transformer-M model.

**Baselines.** We comprehensively compare our Transformer-M with both pre-training methods and 3D molecular models. First, we follow Jiao et al. (2022) to compare several pre-training methods. Hu et al. (2019) proposed a strategy to pre-train GNNs via both node-level and graph-level tasks. Sun et al. (2019) maximized the mutual information between graph-level representations and substructure representations as the pre-training tasks. You et al. (2020) instead used contrastive learning to pre-train GNNs. There are also several works that utilize 3D geometric structures during pre-training. Jing et al. (2021) maximized the mutual information between 2D and 3D representations. Fang et al. (2021) proposed a strategy to learn spatial information by utilizing both local and global 3D structures. Stärk et al. (2022) used two encoders to capture 2D and 3D structural information separately while maximizing the mutual information between 2D and 3D representations. Jiao et al. (2022) adopted an equivariant energy-based model and developed a node-level pretraining loss for force prediction. We report the results of these methods from (Jiao et al., 2022) for comparison.

Second, we follow Thölke & De Fabritiis (2021) to compare 3D molecular models. Schütt et al. (2017) used continuous-filter convolution layers to model quantum interactions in molecules. Anderson et al. (2019) developed a GNN model equipped with activation functions being covariant to rotations. Klicpera et al. (2020) proposed directional message passing, which uses atom-pair embeddings and utilizes directional information between atoms. Schütt et al. (2021) proposed the polarizable atom interaction neural network (PaiNN) that uses an equivariant message passing mechanism. Hutchinson et al. (2021) built upon the Transformer model consisting of attention layers that are equivariant to arbitrary Lie groups and their discrete subgroups. Thölke & De Fabritiis (2021) also developed a Transformer variant with layers designed by prior physical and chemical knowledge. Satorras et al. (2021) proposed the EGNN model which does not require computationally expensive higher-order representations in immediate layers to keep equivariance, and can be easily scaled to higher-dimensional spaces. Godwin et al. (2022) proposed the 3D position denoising task and verified it on the Graph Network-based Simulator (GNS) model (Sanchez-Gonzalez et al., 2020).

**Settings.** We fine-tune the pre-trained Transformer-M on the QM9 dataset. Following Thölke & De Fabritiis (2021), we adopt the Mean Squared Error (MSE) loss during training and use the Mean Absolute Error (MAE) loss function during evaluation. We also adopt label standardization for stable training. We use AdamW as the optimizer, and set the hyper-parameter $\epsilon$ to 1e-8 and $(\beta_1, \beta_2)$ to (0.9,0.999). The gradient clip norm is set to 5.0. The peak learning rate is set to 7e-5. The batch size is set to 128. The dropout ratios for the input embeddings, attention matrices, and hidden

representations are set to 0.0, 0.1, and 0.0 respectively. The weight decay is set to 0.0. The model is fine-tuned for 600k steps with a 60k-step warm-up stage. After the warm-up stage, the learning rate decays linearly to zero. All models are trained on 1 NVIDIA A100 GPU.

## B.5 MORE ANALYSIS

**Investigation on the generality of the design methodology of Transformer-M.** In this work, we develop our Transformer-M model based on the Transformer backbone and integrate separate 2D and 3D channels (implemented by encoding methods) to encode the structural information of 2D and 3D molecular data. As stated in Section 3.2, it is a general design methodology for handling molecular data in different forms, which works well with different structural encoding instantiations. To demonstrate its generality and effectiveness, we further conduct experiments with other structural encodings from GRPE (Park et al., 2022) and EGT (Hussain et al., 2022), which are competitive baselines on the PCQM4Mv2 benchmark as shown in Table 1. All the hyperparameters are kept the same as the settings in Appendix B.1 for a fair comparison. The results are presented in Table 5. It can be easily seen that our Transformer-M model equipped with different encoding methods can consistently obtain significantly better performance than the corresponding vanilla 2D models, which indeed verifies the generality and effectiveness of the design methodology of Transformer-M.

Table 5: Performance of Transformer-M with different structural encodings on PCQM4Mv2 validation set. The evaluation metric is the Mean Absolute Error (MAE) [eV]. For a fair comparison, all hyperparameters are kept the same as the settings in Appendix B.1.

| Structural Encodings | Training in the vanilla setting (2D only) | Training in the Transformer-M framework | $\Delta$ |
|---|---|---|---|
| Graphormer (Ying et al., 2021a) | 0.0878 | 0.0787 | 10.4% |
| GRPE (Park et al., 2022) | 0.0884 | 0.0798 | 9.7% |
| EGT(Hussain et al., 2022) | 0.0876 | 0.0794 | 9.4% |

Table 6: Ablation study on the impact of 3D conformers calculated by different methods. Experiments are conducted on the PCQM4Mv2 dataset. The evaluation metric is the Mean Absolute Error (MAE) [eV]. For a fair comparison, all hyperparameters are kept the same as the settings in Appendix B.1.

| 2D | 3D Conformer [by RDKit (Landrum, 2016)] | 3D Conformer [by DFT, from PCQM4Mv2 (Hu et al., 2021)] | Valid MAE |
|---|---|---|---|
| ✓ | ✗ | ✗ | 0.0878 |
| ✓ | ✓ | ✗ | 0.0872 |
| ✓ | ✓ | ✓ | 0.0792 |
| ✓ | ✗ | ✓ | 0.0787 |

**Investigation on the impact of 3D conformers calculated by different methods.** Besides the versatility to handle molecules in different formats, our Transformer-M further achieves strong performance on various challenging molecular tasks, as shown in Section 4. On PCQM4Mv2 validation set (2D only), our Transformer-M establishes a new state-of-the-art, which mainly credits to the newly introduced 2D-3D joint training strategy in Section 3.2. The chemical knowledge in the 3D geometric structure can be leveraged during joint training and boost the performance of 2D tasks. Since the benefits of the 3D geometric structure are observed, it is natural to ask how the quality of calculated 3D conformers influences the performance of Transformer-M.

To investigate this question, we additionally use the RDKit (Landrum, 2016) to generate one 3D conformed for each molecule in the training set of PCQM4Mv2. Compared to officially provided DFT-optimized geometric structures, the structures are less costly to obtain by using RDKit while also being less accurate. Thus, each molecule has its 2D molecular graph, 3D conformer calculated by DFT, and 3D conformer calculated by RDKit. Based on such a dataset, we conduct three additional experiments. Firstly, we train our Transformer-M model only using 2D molecular graphs. In this experiment, only the 2D channels are activated. Secondly, we train our Transformer-M model using both 2D molecular graphs (encoded by 2D channels) and 3D conformers generated by RDKit (encoded by 3D channels). Thirdly, we train our Transformer-M model using 2D molecular graphs, 3D conformers generated by RDKit, and 3D conformers calculated by DFT. In this experiment, we

use two sets of 3D channels to separately encode structural information of 3D RDKit conformers and 3D DFT conformers. During training, when a data instance enters 3D or 2D+3D modes, both sets of 3D channels are activated and integrated. For all three experiments, the hyperparameters of Transformer-M are kept the same as the settings in Appendix B.1. The results are presented in Table 6.

We can see that the quality of the 3D conformer matters in the final performance: leveraging 3D conformers generated by RDKit (second line) brings minor gains compared to using 2D molecular graphs only (first line). On the contrary, when leveraging 3D conformers calculated by DFT, the improvement is significant (the last two lines). From the practical view, it will be interesting to investigate the influence of 3D conformers calculated by methods that are more accurate than RDKit while more efficient than DFT, e.g., semiempirical methods (Dral et al., 2016), which we leave as future work.

**Investigation on the effectiveness of Transformer-M pre-training.** We provide additional results on the effectiveness of our model on both PDBBind (2D+3D) and QM9 (3D) downstream datasets.

Firstly, to verify the effectiveness of Transformer-M pre-training on the PDBBind dataset, we further pre-train the Graphormer model (Ying et al., 2021a) on the same PCQM4Mv2 dataset as a competitive pre-trained baseline. Since the Graphormer model can only handle graph data, we only use the 2D molecular graph of each data instance. All hyperparameters are kept the same as the settings in Appendix B.1. The results are presented in Table 7.

Table 7: Investigation on the effectiveness of Transformer-M pre-training on PDBBind core set (version 2016). The evaluation metrics include Pearson's correlation coefficient (R), Mean Absolute Error (MAE), Root-Mean Squared Error (RMSE), and Standard Deviation (SD).

| Method | PDBBind core set | | | |
|---|---|---|---|---|
| | R $\uparrow$ | MAE $\downarrow$ | RMSE $\downarrow$ | SD $\downarrow$ |
| SIGN (Li et al., 2021) (the best baseline in Table 2) | $0.797_{\pm 0.012}$ | $1.027_{\pm 0.025}$ | $1.316_{\pm 0.031}$ | $1.312_{\pm 0.035}$ |
| Graphormer (Ying et al., 2021a) | $0.804_{\pm 0.014}$ | $0.998_{\pm 0.005}$ | $1.285_{\pm 0.010}$ | $1.271_{\pm 0.008}$ |
| Transformer-M (ours) | $\mathbf{0.830}_{\pm 0.011}$ | $\mathbf{0.940}_{\pm 0.006}$ | $\mathbf{1.232}_{\pm 0.013}$ | $\mathbf{1.207}_{\pm 0.007}$ |

We can draw the following conclusions: (1) pre-training is helpful (e.g., 0.797 (R of SIGN model, the best baseline) -> 0.804 (R of pre-trained Graphormer model)); (2) our pre-training method is more significant (e.g., 0.804 -> 0.830), which indeed demonstrate the effectiveness of our framework.

Secondly, we demonstrate that our pre-training strategy helps learn a better Transformer model on downstream QM9 dataset. We conduct two additional experiments on the QM9 dataset. In the first experiment, we train the 3D geometric Transformer model (Transformer-M with 3D channel only) from scratch. In the second experiment, we use the 3D Position Denoising task as the objective to pre-train the 3D geometric Transformer on the PCQM4Mv2 and fine-tune the pre-trained checkpoint on QM9. Due to the time limits and constrained resources, we selected six QM9 targets for comparison. All the hyperparameters of pre-training and fine-tuning are kept the same. The results are presented in Table 8.

Table 8: Investigation on the effectiveness of Transformer-M pre-training on QM9 (Ramakrishnan et al., 2014). The evaluation metric is the Mean Absolute Error (MAE).

| Method | $\epsilon_{HOMO}$ | $\epsilon_{LUMO}$ | $U_0$ | U | H | G |
|---|---|---|---|---|---|---|
| No Pre-training | 26.5 | 23.8 | 14.78 | 14.56 | 14.82 | 14.95 |
| 3D Position Denoising only | 18.8 | 17.6 | 10.26 | 10.18 | 10.33 | 10.27 |
| 2D-3D joint Pre-training + 3D Position Denosing | **17.5** | **16.2** | **9.37** | **9.41** | **9.39** | **9.63** |

It can be easily seen that our pre-training methods consistently and significantly improve the downstream performance on all six tasks, which indeed demonstrates the effectiveness of our general framework for 3D molecular data. We are aware that we achieve competitive rather than SOTA performance compared with baselines (5 best performance out of 12 targets, see Table 3). For U0, U, H, and G, there still exists a performance gap between our Transformer-M and some latest baselines, which use pretty complicated neural architectures. We believe that exploring more model alternatives and leveraging the wisdom in those networks into our Transformer-M will further improve the performance, which we will keep working on.

