# OpenReview forum: "One Transformer Can Understand Both 2D & 3D Molecular Data"
_ICLR.cc/2023/Conference — ICLR 2023 poster_

### Official Review · Reviewer_JC8f · 2022-10-23

**Confidence:** 4
**Correctness:** 3
**Technical Novelty And Significance:** 2
**Empirical Novelty And Significance:** 2
**Recommendation:** 6

**Clarity, Quality, Novelty And Reproducibility:**

The paper is clearly written. I believe the results can be reproduced well. My concern is the novelty, as detailed before.

**Strength And Weaknesses:**


#####Strength

1. The motivation of developing a general-purpose model for molecular tasks is great and might be inspiring to the community.

2. The empirical result of the proposed method is competitive, especially on PCQM4Mv2, where the task is the same as the pretraining objective.

3. This manuscript is well organized and easy to follow.

#####Weaknesses

1. The main concern for this paper is that the novelty is not enough to me. To be specifically, both the 2D and 3D branches are proposed by previous work. The positional encodings are verified to be effective by existing work. This work combines these two branches as a single model, which is not novel in terms of technical contribution.

2. Also, the empirical result on PCQM4Mv2 is good. However, the effectiveness of pretraining Transformer-M has not been demonstrated well, empirically. Specifically, the methods used for comparison in Table 2 (PDBBind) and Table 3 (QM9) do not use the same extra data for pretraining. In this sense, the comparison is not so rigorous and fair to me. Even though, the performance on QM9 is not good, except for the energy-related tasks that are consistent with the pretraining tasks. This cannot support the claim that the proposed Transformer-M is a general-purpose model.


**Summary Of The Paper:**

This paper proposes Transformer-M, a Transformer-based model that can take both 2D and 3D molecular formats as input. It adopts various positional encoding techniques to unify 2D and 3D information into transformer as attention bias terms. This model is claimed to be a general-purpose model for molecular tasks. Experiments on several 2D and 3D molecular tasks have been conducted to evaluate the developed method.

**Summary Of The Review:**

Overall, I think this work is a good start for developing a general-purpose model for molecular tasks. The empirical results can be valuable to the community. However, the novelty of this work is below the standard of ICLR.

#####Questions#####

How does Transformer-M deal with the molecules where there might have multiple shortest path between two atoms?


#####After rebuttal#####

Since most of my original concerns are well addressed, I increased the score from 5 to 6. I highly recommend the authors to include the additional experiments in the revision and to consider the technical details of the method, as in my response below.

---

> ### Author Response · Authors · 2022-11-15
> **Response to Reviewer JC8f [2/2]**
>
> **Regarding the effectiveness of pre-training on PDBBind.**
>
> Thanks for the question. To compare models pre-trained on the same extra data, we conduct additional experiments by pre-training the Graphormer model [1] on the same PCQM4Mv2 dataset and fine-tuning it on the PDBBind dataset. Since the Graphormer model can only handle graph data, we only use the 2D molecular graph of each data instance. The results are presented in the following table.
>
> | Method               | R           | MAE         | RMSE        | SD          |
> | -------------------- | ----------- | ----------- | ----------- | ----------- |
> | Graphormer           | 0.804±0.014 | 0.998±0.005 | 1.285±0.010 | 1.271±0.008 |
> | Transformer-M (ours) | 0.830±0.011 | 0.940±0.006 | 1.232±0.013 | 1.207±0.007 |
>
> **Table 1. Performance comparison of models pre-trained on extra data (PCQM4Mv2) on PDBBind core set.**
>
> From Table 1, we can draw the following conclusions: (1) pre-training is helpful (e.g., 0.797 (R of SIGN model, the best baseline) -> 0.804 (R of pre-trained Graphormer model)); (2) our pre-training method is more significant (e.g., 0.804 -> 0.830), which indeed demonstrate the effectiveness of our framework.
>
> **Regarding the effectiveness of pre-training on QM9.**
>
> First of all, we would like to show that the pre-training strategy helps learn a better Transformer model on QM9. To demonstrate this, we conduct two additional experiments on the QM9 dataset. In the first experiment, we train the 3D geometric Transformer model (Transformer-M with 3D channel only) from scratch. In the second experiment, we use the 3D Position Denoising task as the objective to pre-train the 3D geometric Transformer on the PCQM4Mv2 and fine-tune the pre-trained checkpoint on QM9. Due to the time limits and constrained resources, we selected six QM9 targets for comparison. All the hyperparameters of pre-training and fine-tuning are kept the same. The results are presented in the following table.
>
>
> | Method                                           | HOMO | LUMO | U0    | U     | H     | G     |
> | ------------------------------------------------ | ---- | ---- | ----- | ----- | ----- | ----- |
> | No Pre-training                                  | 26.5 | 23.8 | 14.78 | 14.56 | 14.82 | 14.95 |
> | 3D Position Denoising only                       | 18.8 | 17.6 | 10.26 | 10.18 | 10.33 | 10.27 |
> | 2D-3D Joint Pre-training + 3D Position Denoising | 17.5 | 16.2 | 9.37  | 9.41  | 9.39  | 9.63  |
>
> **Table 2. Ablation Study on the impact of pre-training objectives on QM9.**
>
> From Table 2, we can see that our pre-training methods consistently and significantly improve the downstream performance on all six tasks, which indeed demonstrate the effectiveness of our general framework for 3D molecular data.
>
> Second. we are aware that we achieve competitive rather than SOTA performance compared with baselines (5 best performance out of 12 targets). For U0, U, H, and G, there still exists a performance gap between our Transformer-M and some latest baselines, which use pretty complicated neural architectures. We believe that exploring more model alternatives and leveraging the wisdom in those networks into our Transformer-M will further improve the performance, which we will keep working on.
>
>
> **Regarding the shortest path calculation.**
>
> It is a good question. In our work, we randomly pick one shortest path (SP) if there are multiple SPs. To better understand whether the sampling is a big issue, we sampled a subset of graphs in the PCQM4Mv2 dataset (10K molecular graphs) and counted the number of SPs between all node pairs for each graph. We find that about 90\% of node pairs have only one SP, about 10\% have two SPs. Interestingly, we find that for most node pairs with two SPs, the two SPs are symmetric, e.g., two carbon atoms opposite each other on the benzene ring. So randomly picking any SP would give the same representation and never hurt the model training/inference.
>
> [1] Ying, Chengxuan, et al. "Do transformers really perform badly for graph representation?." Advances in Neural Information Processing Systems 34 (2021): 28877-28888.
>
> ***** May you have any further questions, please tell us and we are willing to address your concerns. *****

---

> > ### Author Response · Authors · 2022-11-17
> > **Response to Reviewer JC8f - Looking forward to your feedback**
> >
> > Thanks again for your efforts in reviewing our paper. We have updated our paper by adding new results and discussions as you advised. We sincerely hope that the reviewers can re-evaluate our submission based on our responses and revision. We are also willing to have more discussions if the reviewers have further questions.

---

> > ### Comment · Reviewer_JC8f · 2022-11-17
> > **Response**
> >
> > Thank you for the response and clarification.
> >
> > For the added pre-training experiments on PDBBind, (1) what is the performance for Graphormer without pre-training? (2) The comparison is till not rigorous since the baseline methods do not use 3D information during pre-training as Transformer-M.
> >
> > In terms of the shortest path consideration, why don't we consider all shortest path to avoid the ambiguity.

---

> > > ### Author Response · Authors · 2022-11-18
> > > **Response to Reviewer JC8f**
> > >
> > > Thanks for the feedback! Following your suggestion, we quickly conducted additional experiments on PDBBind. The results include the performance of 1) Graphormer [1] without pre-training; 2) 3D geometric Transformer [2] without pre-training; 3) 3D geometric Transformer with pre-training (the same 3D data used in Transformer-M pre-training).
> > >
> > > | Method               | R           | MAE         | RMSE        | SD          |
> > > | -------------------- | ----------- | ----------- | ----------- | ----------- |
> > > | Graphormer w/o pre-training                   | 0.773±0.021 | 1.088±0.017 | 1.396±0.023 | 1.389±0.028 |
> > > | Graphormer with pre-training                  | 0.804±0.014 | 0.998±0.005 | 1.285±0.010 | 1.271±0.008 |
> > > | 3D geometric Transformer w/o pre-training     | 0.782±0.019 | 1.063±0.022 | 1.359±0.026 | 1.351±0.031 |
> > > | 3D geometric Transformer with pre-training    | 0.819±0.014 | 0.967±0.008 | 1.260±0.013 | 1.233±0.009 |
> > > | Transformer-M (ours) | 0.830±0.011 | 0.940±0.006 | 1.232±0.013 | 1.207±0.007 |
> > >
> > > **Table 1. Performance comparison of models pre-trained or not on extra data (PCQM4Mv2) evaluated on PDBBind core set.**
> > >
> > > From the above results, we can see that (1). the pre-training is helpful. Given any architecture choice, fine-tuning from a pre-trained checkpoint is consistently better than training the model from scratch; (2). Compared among all pre-trained checkpoints (line 2,4,5), our Transformer-M performs best. We believe this comparison is fair and convincing to demonstrate the effectiveness of our framework.
> > >
> > > In terms of the calculation of the shortest path, we strictly follow the original Graphormer [1] to select one shortest path (SP) for a fair comparison. Note that using one sampled shortest path will not bring significant ambiguity for molecular domain. As we show in the first round of rebuttal, only about 10\% of instances have two SPs. For most node pairs with two SPs, the two SPs are symmetric, e.g., two carbon atoms opposite each other on the benzene ring. So randomly picking any SP or using them both by averaging would give the same representation.
> > >
> > > We agree with you that multiple different shortest paths between nodes may exist in other graph domains, such as social networks. We think using multiple SPs is the right choice and will investigate the empirical performance of using multiple SPs in the future.
> > >
> > > Thanks again for your efforts in reviewing our paper. We sincerely hope the above results and discussions can address your concerns and help you reevaluate our work.
> > >
> > > [1] Ying, Chengxuan, et al. "Do transformers really perform badly for graph representation?." Advances in Neural Information Processing Systems 34 (2021): 28877-28888.
> > > [2] Shi, Yu, et al. "Benchmarking graphormer on large-scale molecular modeling datasets." arXiv preprint arXiv:2203.04810 (2022).

---

> > > > ### Comment · Reviewer_JC8f · 2022-11-18
> > > > **Thanks**
> > > >
> > > > Thanks for the response. Since the pre-training experiments are added, I would like to increase the score. I highly recommend adding these results to the main paper.
> > > >
> > > > For the shortest path issue, I still think that a more technically sound solution should be considered in the final version or future work.

---

> > > > > ### Author Response · Authors · 2022-11-19
> > > > > **Sincerely thank you for your feedback!**
> > > > >
> > > > > Thanks again for your efforts in reviewing our paper! We will carefully reorganize our paper to add the advised experimental results to the main body of our paper in the final version.  We will also keep investigating the ways to use the shortest path information.
> > > > >
> > > > > Best regards,
> > > > >
> > > > > Paper 452 Authors

---

> ### Author Response · Authors · 2022-11-15
> **Response to Reviewer JC8f [1/2]**
>
> Thank you for spending time reviewing our paper. Here are our responses to your questions:
>
> **Regarding the novelty of our work.**
>
> We respectfully disagree with Reviewer JC8f that our contribution is limited. We agree that all structural encodings used in this paper have been verified in previous works. However, we have already given credits to these methods in the related work section and never highlighted them as a part of the contribution.
>
> We want to clarify that our contribution is creating a path toward learning a single model for both 2D and 3D molecular data. We are the first to tackle this problem and figure out how to achieve it. Along this path, we make a series of key contributions:
> - Firstly, we develop the backbone model. Instead of focusing on sophisticated modules specialized for only one molecular data form, we establish a *general design methodology* (Transformer integrated with separate 2D and 3D channels) to unify both 2D and 3D structural information of molecules;
> - Furthermore, we introduce *effective training strategies* (2D/3D/2D+3D modes) for this unified model. Despite the simplicity, our introduced training strategies can utilize both unpaired and paired 2D and 3D data and are compatible with different kinds of training objectives, which indeed bring general adaptability into our framework;
> - We also develop *proper training objectives* (2D-3D Joint Pre-training and 3D Position Denoising), which unleash the power of our model to achieve strong performance on molecular tasks in different forms.
>
> We believe this whole framework is novel to the community, which shows this direction (unified molecular modelling) is feasible and promising. From this perspective, our work is original.
>
> We would like to thank the reviewer, as this concern helps us realize our writing problems. In the submitted version, we put much effort into describing the architecture details but ignored highlighting the training part. We will make our contribution clear in the new version of the paper.

---

### Official Review · Reviewer_yHnH · 2022-10-24

**Confidence:** 4
**Correctness:** 4
**Technical Novelty And Significance:** 4
**Empirical Novelty And Significance:** 3
**Recommendation:** 8

**Clarity, Quality, Novelty And Reproducibility:**

The work is rather original in combining 2D or 3D information of small molecules at will in a transformer.  The quality of the work is high: it builds on top of other existing state-of-the-art ideas, and yet it leaves room for future improvement.  The paper is clear, and the main point of this work is rather simple and powerful. The paper comes with code (though I didn't try to test it myself).


**Strength And Weaknesses:**

A core strength of this paper is that the high-level idea is simple and clean, yet the results show that it works well: Transformer-M ranks top on the small-molecule OpenGraph large-scale challenge, a world-wide public competition challenge.

This work is rather strong.  The paper starts to explain the rationale for why the method works as well as it does by performing an ablation study.  I'd have liked to see a couple additional ablations, or other ways to explain why the method works so well: can one keep 3D position denoising for the 3D entries, but drop the joint pre-trainig? How strongly do the various specific features in the pair-wise channels influence the final result?  Do we expect the 2D and 3D features to be "aligned" for a given molecule after pre-training the model? Finally, and only as a minor curiosity, what is the impact, if any, of the mode distribution on a downstream task like the QM9 (and if you had that info, could you possibly combine tables 4 and 5 to a single one?)


**Summary Of The Paper:**

This paper documents a simple method for combining the input representations of a small molecule in 2D and 3D in a single neural network model. The work demonstrates how one can train models using subsets of 2D, 3D, or mixed representations, by modifying the attention head of a transformer architecture to explicitly add a sum over channels of pairwise features generated from (2D) small molecule graphs and channels of pairwise features generated from the small molecule coordinates in 3D. The authors combine a small number of concepts from previous state-of-the art models for small molecules in selecting these features and the unsupervised pre-training tasks, and they manage to show that their new architecture achieves peak performance in the opengraph large-scale challenge for small molecules, and decent performance throughout other tasks.


**Summary Of The Review:**

This paper ranks top in a rather important small-molecule benchmark; the idea is simple and powerful and the implementation is straightforward.  I think it deserves publication at ICLR2023.

---

> ### Author Response · Authors · 2022-11-15
> **Response to Reviewer yHnH [2/2]**
>
> **Regarding the 2D and 3D feature alignment.**
>
> It is a good catch! We hope that the 2D and 3D features can be aligned well, which suggests that rich chemical knowledge has been transferred across different data formats. Our Transformer-M model implicitly encourages such property through joint training, which we show has significantly improved downstream task performance. We also observed that the loss of the 2D mode is a bit higher than that of the 3D mode, showing that the 2D and 3D features still have some differences. This observation deserves further efforts to explore, which we will keep working on in the future.
>
> **Regarding the impact of pre-training objectives and mode distribution on downstream tasks.**
> Thanks for the suggestion! We further conduct additional experiments on the impact of pre-training objectives and mode distribution on downstream tasks. As you suggested, we merge the results from Table 4 and Table 5 in our paper to the following table.
>
> | 2D-3D joint Pre-training | 3D Position Denoising | Mode Distribution | PCQM4Mv2 Valid MAE | HOMO (QM9) | LUMO (QM9) | Gap(QM9) |
> | ------------------------ | --------------------- | ----------------- | ------------------ | ---------- | ---------- | -------- |
> | ×                        | ×                     | -                 | 0.0878             | 26.5       | 23.8       | 42.3     |
> | √                        | ×                     | 1:2:1             | 0.0811             | 20.1       | 19.3       | 31.2     |
> | √                        | √                     | 1:2:1             | 0.0787             | 17.5       | 16.2       | 27.4     |
> | √                        | √                     | 1:1:1             | 0.0796             | 18.1       | 16.9       | 27.9     |
> | √                        | √                     | 1:2:2             | 0.0789             | 17.8       | 16.6       | 27.5     |
>
> **Table 2. Impact of pre-training tasks and mode distribution on the performance of our Transformer-M model.**
>
> From Table 2, it can be easily seen that our pre-training objectives (2D-3D joint pre-training and 3D Position Denoising) and strategies (2D/3D/2D+3D modes) indeed boost the performance of different downstream tasks. Besides, the 2D-3D joint pre-training is robust to the mode distribution on both tasks. Overall, the results further demonstrate the effectiveness of our framework.
>
>
>
> [1] Park, Wonpyo, et al. "GRPE: Relative Positional Encoding for Graph Transformer." ICLR2022 Machine Learning for Drug Discovery. 2022.
>
> [2] Hussain, Md Shamim, Mohammed J. Zaki, and Dharmashankar Subramanian. "Global self-attention as a replacement for graph convolution." Proceedings of the 28th ACM SIGKDD Conference on Knowledge Discovery and Data Mining. 2022.
>
> [3] Ying, Chengxuan, et al. "Do transformers really perform badly for graph representation?." Advances in Neural Information Processing Systems 34 (2021): 28877-28888.
>
> ***** May you have any further questions, please tell us and we are willing to address your concerns. *****

---

> ### Author Response · Authors · 2022-11-15
> **Response to Reviewer yHnH [1/2]**
>
> Thank you very much for supporting our work! We appreciate your advice on the experiments. Here are our responses to your questions:
>
> **Regarding only using 3D Position Denoising to pre-train the Transformer-M model.**
>
> Thanks for the question. We have already demonstrated in the paper that 3D Position Denoising task is quite helpful in training our Transformer-M model. However, purely relying on this pre-training task cannot improve 2D downstream tasks as the model never observes 2D information during the pre-training stage, which we have verified empirically.
>
> **Regarding the ablation experiments on the pair-wise channels.**
>
> Thanks for the suggestion! It is a good catch that our general framework can use different instantiations of pair-wise channels. Following your advice, we further experiment with other pair-wise structural encodings in GRPE [1] and EGT [2]. All the hyperparameters are kept the same as the settings in Appendix B.1. The results are presented in the following table.
>
> | Structural Encodings | Training in the vanilla setting (2D only) | Training in the Transformer-M framework | Relative Improvements |
> | -------------------- | ----------------------------------------- | --------------------------------------- | --------------------- |
> | Graphormer [3]       | 0.0878                                    | 0.0787                                  | 10.4%                 |
> | GRPE [1]             | 0.0884                                    | 0.0798                                  | 9.7%                  |
> | EGT  [2]             | 0.0876                                    | 0.0794                                  | 9.4%                  |
>
> **Table 1. Performance of Transformer-M using different structural encodings on PCQM4Mv2 validation set.**
>
> From Table 1, it can be easily seen that our Transformer-M (unified model) equipped with different encoding methods can consistently obtain significantly better performance than the corresponding vanilla 2D models, which further verifies the generality and effectiveness of our Transformer-M model.

---

### Official Review · Reviewer_8vsr · 2022-10-25

**Confidence:** 4
**Correctness:** 3
**Technical Novelty And Significance:** 2
**Empirical Novelty And Significance:** 2
**Recommendation:** 5

**Clarity, Quality, Novelty And Reproducibility:**

Clarity: The paper is reader friendly.
Quality and Novelty: Not very novel. It looks like a simple combination of previous works.
Reproducibility: The code will be released. But I don't have much confidence in the reproduction of the results on PDBBind.


**Strength And Weaknesses:**

Strengths

• The paper is generally well-written and easy to follow.

• Transformer-M can encode 2D or 3D structural information as bias terms and add to the attention matrix. It also encodes atomic centrality and adds to the atom features. Based on this, the model can obtain molecular representations from different data modes.

• During pretraining, the authors label the data with different data modes for joint training. This training strategy may improve the performance of Transformer-M on downstream tasks from the results.

Weaknesses

• 2D and 3D information encoding are from previous work. This work just simply combines them, and the model architecture is the same as Graphormer[1]. The novelty is not enough.

• Supervised pretraining based on the prediction of homo-lumo gap may lead to negative transfer. For example, on QM9 in downstream experiments, Transformer-M performs poorly on most tasks other than homo, lumo, and gap. This may be contradictory to the description "general-purpose neural network model" claimed in this paper.

• Lack of description of PDBBind data processing and splitting in downstream tasks.

• Absence of some ablation experiments: ①(p2D, p3D, p2D&3D) = (1:0:0) / (0:1:0) / (0:0:1); ②Only using the 3D position denoising task while pretraining.

Other questions：

• Do the authors consider encoding 1D molecular data mode, e.g., SMILES,  simultaneously?

• What do the authors think about the possibility of negative transfer on downstream tasks due to the supervised signal introduced during pretraining?

• Whether there is data leakage during finetuning on PDBBind, because we know that the general, refined, and core sets have overlapping parts.


**Summary Of The Paper:**

Molecules can be represented in a variety of chemical data modes, such as a 2D graph or a collection of atoms in 3D space. Most previous work in molecular representation learning has designed networks for a specific data mode, with the risk of failing to learn from other modes. The authors argue that a neural network that is chemically generalized should be able to handle molecule-related tasks across data modes. To accomplish this, the authors created Transformer-M, which is based on the Transformer and can be fed with 2D or 3D molecular data. The results of experiments indicate that Transformer-M performs well in 2D, 3D, and 2D&3D tasks.

**Summary Of The Review:**

In summary, this paper is reader-friendly, but novelty is not enough, and some descriptions of the experiments are not clear enough.

---

> ### Author Response · Authors · 2022-11-15
> **Response to Reviewer 8vsr [3/3]**
>
> **Regarding the 1D molecular data mode.**
>
> It is a good question. We purposely do not take the 1D format into consideration as the 1D format (e.g., SMILES) and the 2D format can be converted between each other without expensive costs. Thus, if molecular data comes in 1D format, we can conveniently transform it into the 2D format and feed it into Transformer-M. The transformation between 2D and 3D-format data is sometimes computationally expensive, especially when you require an accurate 3D conformer.
>
> **Regarding whether there exists negative transfer on the performance of downstream tasks.**
>
> The negative transfer usually refers to the phenomenon that fine-tuning a pre-trained checkpoint leads to worse performance than training the model from scratch. Our work is not the case.
>
> To show this, we conduct two additional experiments on the QM9 dataset. In the first experiment, we train the 3D geometric Transformer model (Transformer-M with 3D channel only) from scratch. In the second experiment, we use the 3D Position Denoising task as the objective to pre-train the 3D geometric Transformer on the PCQM4Mv2 and fine-tune the pre-trained checkpoint on QM9. Due to the time limits and constrained resources, we select six QM9 targets for comparison. All the hyperparameters of pre-training and fine-tuning are kept the same. The results are presented in the following table.
>
>
> | Method                                           | HOMO | LUMO | U0    | U     | H     | G     |
> | ------------------------------------------------ | ---- | ---- | ----- | ----- | ----- | ----- |
> | No Pre-training                                  | 26.5 | 23.8 | 14.78 | 14.56 | 14.82 | 14.95 |
> | 3D Position Denoising only                       | 18.8 | 17.6 | 10.26 | 10.18 | 10.33 | 10.27 |
> | 2D-3D Joint Pre-training + 3D Position Denoising | 17.5 | 16.2 | 9.37  | 9.41  | 9.39  | 9.63  |
>
> **Table 2. Ablation Study on the impact of pre-training objectives on QM9.**
>
> From Table 2, we can see that our pre-training methods significantly improved the downstream performance on all six tasks, and the negative transfer didn't happen.
>
> For the performance of our model on QM9, we are aware that we achieve competitive rather than superior performance compared with baselines (5 best performance out of 12 targets). For U0, U, H, and G, there still exists a performance gap between our Transformer-M and some latest baselines, which use pretty complicated neural architectures. We believe that exploring more model alternatives and leveraging the wisdom in those networks into our Transformer-M will further improve the performance, which we will keep working on in the future.
>
> [1] Li, Shuangli, et al. "Structure-aware interactive graph neural networks for the prediction of protein-ligand binding affinity." Proceedings of the 27th ACM SIGKDD Conference on Knowledge Discovery & Data Mining. 2021.
>
> ***** May you have any further questions, please tell us and we are willing to address your concerns. *****

---

> > ### Author Response · Authors · 2022-11-17
> > **Response to Reviewer 8vsr - Looking forward to your feedback**
> >
> > Thanks again for your efforts in reviewing our paper. We have updated our paper by adding new results and discussions as you advised. We sincerely hope that the reviewers can re-evaluate our submission based on our responses and revision. We are also willing to have more discussions if the reviewers have further questions.

---

> > > ### Comment · Reviewer_8vsr · 2022-11-18
> > > **Thank you for the response.**
> > >
> > > Thank you for your detailed response. There appears to be no data leakage, and the results are quite impressive. I am willing to raise my score, but  I still have a few questions about the PDBBind task. Since baseline models such as SIGN are not pretrained, this comparison appears unfair. It seems more reasonable to benchmark Transformer-M pretraining on PDBBind.

---

> > > > ### Author Response · Authors · 2022-11-18
> > > > **Response to Reviewer 8vsr**
> > > >
> > > > Thanks for the feedback! Both you and Reviewer JC8f have the same question about benchmarking Transformer-M pretraining on PDBBind. We have quickly conducted additional experiments. The results include the performance of 1) Graphormer [1] without pre-training; 2) Graphormer with pre-training (the same 2D data used in Transformer-M pre-training); 3) 3D geometric Transformer [2] without pre-training; 4) 3D geometric Transformer with pre-training (the same 3D data used in Transformer-M pre-training).
> > > >
> > > > | Method               | R           | MAE         | RMSE        | SD          |
> > > > | -------------------- | ----------- | ----------- | ----------- | ----------- |
> > > > | Graphormer w/o pre-training                   | 0.773±0.021 | 1.088±0.017 | 1.396±0.023 | 1.389±0.028 |
> > > > | Graphormer with pre-training                  | 0.804±0.014 | 0.998±0.005 | 1.285±0.010 | 1.271±0.008 |
> > > > | 3D geometric Transformer w/o pre-training     | 0.782±0.019 | 1.063±0.022 | 1.359±0.026 | 1.351±0.031 |
> > > > | 3D geometric Transformer with pre-training    | 0.819±0.014 | 0.967±0.008 | 1.260±0.013 | 1.233±0.009 |
> > > > | Transformer-M (ours) | 0.830±0.011 | 0.940±0.006 | 1.232±0.013 | 1.207±0.007 |
> > > >
> > > > **Table 1. Performance comparison of models pre-trained or not on extra data (PCQM4Mv2) evaluated on PDBBind core set.**
> > > >
> > > > From the above results, we can see that (1). the pre-training is helpful. Given any architecture choice, fine-tuning from a pre-trained checkpoint is consistently better than training the model from scratch; (2). Compared among all pre-trained checkpoints (line 2,4,5), our Transformer-M performs best. We believe this comparison is fair and convincing to demonstrate the effectiveness of our framework.
> > > >
> > > > Thanks again for your efforts in reviewing our paper. We sincerely hope the above results and discussions can address your concerns and help you reevaluate our work.
> > > >
> > > > [1] Ying, Chengxuan, et al. "Do transformers really perform badly for graph representation?." Advances in Neural Information Processing Systems 34 (2021): 28877-28888.
> > > >
> > > > [2] Shi, Yu, et al. "Benchmarking graphormer on large-scale molecular modeling datasets." arXiv preprint arXiv:2203.04810 (2022).

---

> > > > > ### Author Response · Authors · 2022-11-20
> > > > > **More feedback**
> > > > >
> > > > > Thanks again for your efforts in reviewing our paper. We sincerely appreciate your reevaluation and increasing the score on our paper. However, unlike the other three reviewers, who all give positive scores, it seems that you still feel the work is below acceptance without any further reasons.
> > > > >
> > > > > In the above responses, we believe that most of your concerns have been addressed, including:
> > > > > - the novelty of our work;
> > > > > - the details of PDBBind experiments and the data leakage concern;
> > > > > - the additional ablation experiments;
> > > > > - the 1D molecular data mode;
> > > > > - the negative transfer issue;
> > > > > - benchmarking Transformer-M pre-training on PDBBind.
> > > > >
> > > > > We have also revised our paper according to your suggestions.
> > > > >
> > > > > It would be great if you could tell us your remaining concerns, which will help us improve the quality of the paper and meet your expectation. Looking forward to your feedback!

---

> ### Author Response · Authors · 2022-11-15
> **Response to Reviewer 8vsr [2/3]**
>
> **Regarding the details of PDBBind experiments and the data leakage concern.**
>
> Thanks for the suggestion! We further provide the details of PDBBind datasets. For a fair comparison, we follow [1] to use the PDBBind v2016 dataset, which indeed includes three overlapping subsets, i.e. general, refined, and core set. The general set contains all 13,283 protein-ligand complexes, while the 4,057 complexes in the refined set are selected out of the general set with better quality. Moreover, the core set with 290 complexes serves as the highest quality benchmark for testing. To avoid data leakage, we remove the data instances in the core set from the refined set. Thus, we obtain 3,767 complexes as the training set. After training, we evaluate our model on the core set, which contains 290 complexes.
>
> **Regarding the ablation experiments.**
>
> Thanks for the suggestion! We further provide additional results on the impact of task probabilities on our Transformer-M model. In detail, we set the task probabilities ($p_{2D}$, $p_{3D}$, $p_{2D-3D}$) of each data instance during training to 1) (1.0, 0.0, 0.0); 2) (0.0, 1.0, 0.0); 3) (0.0, 0.0, 1.0); 4) (1/3, 1/3, 1/3); 5) (0.2, 0.4, 0.4); 6) (0.2, 0.6, 0.2). All other hyperparameters of Transformer-M are kept the same as the settings in Appendix B.1. The results are presented in the following table.
>
> | Task Probabilities ($p_{2D}$, $p_{3D}$, $p_{2D-3D}$) | Valid MAE on PCQM4Mv2 |
> | ---------------------------------------------------- | --------------------- |
> | (1, 0, 0)                                            | 0.0878                |
> | (0, 1, 0)                                            | 0.6219                |
> | (0, 0, 1)                                            | 0.3178                |
> | (1/3, 1/3, 1/3)                                      | 0.0796                |
> | (0.2, 0.4, 0.4)                                      | 0.0789                |
> | (0.2, 0.6, 0.2)                                      | 0.0787                |
>
> **Table 1. Additional results on the impact of task probabilities on the Transformer-M model.**
>
> From Table 1, we can see that, firstly, only using one mode to train the Transformer-M model will not help PCQM4Mv2 task performance. For example, the performance of the models trained only using 2D+3D(or 3D) mode is significantly worse than that of the 2D-3D joint training setting. It is because the model trained in this way always leverages the 3D structural signal during training, which makes the model perform worse in downstream 2D tasks when 3D structural information is absent.
>
> Secondly, 2D-3D joint training is robust and leads to better results. We can see that when we slightly change the task probabilities around the uniform distribution, the learned models always bring significant performance gains. This observation indicates that large parts of the gains are attributed to our training method, and the results are reliable.

---

> ### Author Response · Authors · 2022-11-15
> **Response to Reviewer 8vsr [1/3]**
>
> Thank you for spending time reviewing our paper. Here are our responses to your questions:
>
> **Regarding the novelty of our work.**
>
> We respectfully disagree with Reviewer 8vsr that our contribution is limited. We agree that all structural encodings used in this paper have been verified in previous works. However, we have already given credits to these methods in the related work section and never highlighted them as a part of the contribution.
>
> We want to clarify that our contribution is creating a path toward learning a single model for both 2D and 3D molecular data. We are the first to tackle this problem and figure out how to achieve it. Along this path, we make a series of key contributions:
> - Firstly, we develop the backbone model. Instead of focusing on sophisticated modules specialized for only one molecular data form, we establish a *general design methodology* (Transformer integrated with separate 2D and 3D channels) to unify both 2D and 3D structural information of molecules;
> - Furthermore, we introduce *effective training strategies* (2D/3D/2D+3D modes) for this unified model. Despite the simplicity, our introduced training strategies can utilize both unpaired and paired 2D and 3D data and are compatible with different kinds of training objectives, which indeed bring general adaptability into our framework;
> - We also develop *proper training objectives* (2D-3D Joint Pre-training and 3D Position Denoising), which unleash the power of our model to achieve strong performance on molecular tasks in different forms.
>
> We believe this whole framework is novel to the community, which shows that the direction (unified molecular modelling) is feasible and promising. From this perspective, our work is original.
>
> We would like to thank the reviewer, as this concern helps us realize our writing problems. In the submitted version, we put much effort into describing the architecture details but ignored highlighting the training part. We will make our contribution clear in the new version of the paper.

---

### Official Review · Reviewer_do3S · 2022-10-29

**Confidence:** 5
**Clarity, Quality, Novelty And Reproducibility:** The paper is well written and easy to…
**Correctness:** 3
**Technical Novelty And Significance:** 2
**Empirical Novelty And Significance:** 3
**Recommendation:** 6

**Strength And Weaknesses:**

The modifications are only a few lines of code over the original Graphormer 3D proposed in [1]. Still, this simple modification results in a new STOTA on PCQM4Mv2, namely introducing a mask in the input based on whether the network should operate in 2D/3D/Both.

The experiments on the task probabilities, which is the primary modification, are limited; changing this ratio can drastically change the model's performance. Most of the gains are attributed to the pretraining task and 3D encoding in the bias term, which was previously leveraged in [1].

I would have liked to see more experiments on trying to further leverage 3D information on 2D inputs that don't have the position information, for example, adding conformers as well. Would the network operated in 3D/2D mode perform the best over simply 2D?
What happens if you generate multiple conformers and switch these modes on and off? Would there be a gain in performance? The authors could have done more experiments on this front.

**Summary Of The Paper:**

The authors propose a simple modifications over the Graphormer 3D [1] , which previously encoded node, graph level information through additional biases added to the self-attention modules, namely node-encoding via in/out degree embeddings, path encoding via shortest path and edge embedding aggregation along this path. To encode the spatial and centrality information for 3D molecules they embed these using a GBF assuming a fully connected graph as opposed to using cutoffs.

Now, the authors simply introduce a switching mechanism in this model, where the corresponding 2D/3D encodings are switched on and off.  This is done as it may be the case that 2D information is available and 3D information is not on a particular input sample, so to allow the model to distill "3D" level information to enrich the 2D representation, for example it is the case for the PCQM4Mv2 dataset which includes a highly optimized conformer for the train-set but not the rest.



[1] Shi, Yu, et al. "Benchmarking graphormer on large-scale molecular modeling datasets." arXiv preprint arXiv:2203.04810 (2022).

**Summary Of The Review:**

To summarize it's a simple idea that works, and the empirical results are good.

---

> ### Author Response · Authors · 2022-11-15
> **Response to Reviewer do3S [2/2]**
>
> **Regarding the experiments on molecules with different conformer information.**
>
> Thanks for the suggestion! We further conduct experiments to investigate the influence of 3D conformers calculated by different methods. For each molecule in the training set of PCQM4Mv2, we additionally use the RDKit [1] to generate one 3D conformer. Thus, each molecule has its 2D molecular graph, 3D conformer calculated by DFT, and 3D conformer calculated by RDKit. Based on such dataset, we conduct three additional experiments. Firstly, we train our Transformer-M model only using 2D molecular graphs. In this experiment, only the 2D channels are activated. Secondly, we train our Transformer-M model using both 2D molecular graphs (encoded by 2D channels) and 3D conformers generated by RDKit (encoded by 3D channels). Thirdly, we train our Transformer-M model using 2D molecular graphs, 3D conformers generated by RDKit, and 3D conformers calculated by DFT. In this experiment, we use two sets of 3D channels to separately encode structural information of 3D RDKit conformers and 3D DFT conformers. During training, when a data instance enters 3D or 2D+3D modes, both sets of 3D channels are activated and integrated. For all three experiments, the hyperparameters of Transformer-M are kept the same as the settings in Appendix B.1. The results are presented in the following table.
>
> | 2D   | 3D Conformer (RDKit) | 3D Conformer (DFT, from PCQM4Mv2) | Valid MAE on PCQM4Mv2 |
> | ---- | -------------------- | --------------------------------- | --------------------- |
> | √    | ×                    | ×                                 | 0.0878                |
> | √    | √                    | ×                                 | 0.0872                |
> | √    | √                    | √                                 | 0.0792                |
> | √    | ×                    | √                                 | 0.0787                |
>
> **Table 2. Ablation study on the impact of 3D conformers calculated by different methods.**
>
> We can see that the quality of the 3D conformer matters in the final performance: Leveraging 3D conformers generated by RDKit (second line) brings minor gains compared to using 2D molecular graphs only (first line). On the contrary, when leveraging 3D conformers calculated by DFT, the improvement is significant (the last two lines). From the practical view, it will be interesting to investigate the influence of 3D conformers calculated by methods that are more accurate than RDKit while more efficient than DFT, e.g., semiempirical methods [2]. Due to the time limits and constrained resources, we leave it as future work.
>
> [1] RDKit: Open-Source Cheminformatics Software. https://www.rdkit.org/
>
> [2] Dral, Pavlo O., et al. "Semiempirical quantum-chemical orthogonalization-corrected methods: benchmarks for ground-state properties." Journal of Chemical Theory and Computation 12.3 (2016): 1097-1120.
>
> ***** May you have any further questions, please tell us and we are willing to address your concerns. *****

---

> > ### Author Response · Authors · 2022-11-17
> > **Response to Reviewer do3S - Looking forward to your feedback**
> >
> > Thanks again for your efforts in reviewing our paper. We have updated our paper by adding new results and discussions as you advised. We sincerely hope that the reviewers can re-evaluate our submission based on our responses and revision. We are also willing to have more discussions if the reviewers have further questions.

---

> ### Author Response · Authors · 2022-11-15
> **Response to Reviewer do3S [1/2]**
>
> Thank you very much for supporting our work! We appreciate your advice on the experiments. Here are our responses to your questions:
>
> **Regarding the ablation on the task probabilities and where the performance gain comes from.**
>
> Thanks for the suggestion! We further provide additional results on the impact of task probabilities on our Transformer-M model. In detail, we set the task probabilities ($p_{2D}$, $p_{3D}$, $p_{2D-3D}$) of each data instance during training to 1) (1.0, 0.0, 0.0); 2) (0.0, 1.0, 0.0); 3) (0.0, 0.0, 1.0); 4) (1/3, 1/3, 1/3); 5) (0.2, 0.4, 0.4); 6) (0.2, 0.6, 0.2). All other hyperparameters of Transformer-M are kept the same as the settings in Appendix B.1. The results are presented in the following table.
>
> | Task Probabilities ($p_{2D}$, $p_{3D}$, $p_{2D-3D}$) | Valid MAE on PCQM4Mv2 |
> | ---------------------------------------------------- | --------------------- |
> | (1, 0, 0)                                            | 0.0878                |
> | (0, 1, 0)                                            | 0.6219                |
> | (0, 0, 1)                                            | 0.3178                |
> | (1/3, 1/3, 1/3)                                      | 0.0796                |
> | (0.2, 0.4, 0.4)                                      | 0.0789                |
> | (0.2, 0.6, 0.2)                                      | 0.0787                |
>
> **Table 1. Additional results on the impact of task probabilities on the Transformer-M model.**
>
> From Table 1, we can see that, firstly, only using one mode to train the Transformer-M model will not help PCQM4Mv2 task performance. For example, the performance of the models trained only using 2D+3D(or 3D) mode is significantly worse than that of the 2D-3D joint training setting. It is because the model trained in this way always leverages the 3D structural signal during training, making the model perform worse in downstream 2D tasks when 3D structural information is absent.
>
> Secondly, 2D-3D joint training is robust and leads to better results. We can see that when we change the task probabilities around the uniform distribution, the learned models bring significant and similar performance gains. This observation indicates that a large part of the gains are attributed to our training methods, and the results are reliable.

---

### Author Response · Authors · 2022-11-17
**General Response: New Results & Paper Updates**

We sincerely thank all the reviewers and the area chair for their efforts in reviewing our paper. As you advised, we add new results and discussions to our paper, including:

- **Highlight our key contributions.** [Section 1 and Section 3.2, Reviewer 8vsr and JC8f]
- **Additional results on the impact of mode distribution on downstream tasks.** [Table 4, Section 4.5, Reviewer yHnH]
- **Details of the PDBBind dataset.** [Section B.3 (the Dataset paragraph), Reviewer 8vsr]
- **Discussions on the generality of the design methodology of Transformer-M.** [Table 5, Section B.5, Reviewer yHnH]
- **Discussions on the impact of 3D conformers calculated by different methods.** [Table 6, Section B.5, Reviewer do3S]
- **Discussions on the effectiveness of Transformer-M pre-training.** [Table 7 and 8, Section B.5, Reviewer 8vsr and JC8f]

Please let us know if you have any further concerns and we are willing to answer any further questions you have about our paper. Thank you again for your feedback.

Thanks!

Paper 452 Authors

---

### Public Comment · ~Yoni_Choukroun1 · 2023-06-15
**References**

Thank you for the great work!

Just wanted to highlight a few of the previous related works about Transformer based architectures for molecular data.

Maziarka, Łukasz, et al. "Relative molecule self-attention transformer." arXiv preprint arXiv:2110.05841 (2021).\
Wu, Fang, et al. "3d-transformer: Molecular representation with transformer in 3d space." arXiv preprint arXiv:2110.01191 (2021).\
Choukroun, Yoni, and Lior Wolf. "Geometric transformer for end-to-end molecule properties prediction." arXiv preprint arXiv:2110.13721 (2021).

---

### Decision · Program_Chairs · 2023-01-20

**Decision:**

Accept: poster

**Justification For Why Not Higher Score:**

The work proposed a unified transformer-based framework for 2D and 3D molecular graphs, which is an important contribution to this field. However, the motivations and insights behind the proposed method are not fully discussed, which limits the theoretical contribution of this work.

**Justification For Why Not Lower Score:**

In the rebuttal phase, the reviewers proposed some comments to the submission, most of which are about the experimental part. The authors resolved the comments by adding more experiments. The reviewers are satisfied with the reply, and the scores of the paper are 6, 5, 8, 6. The reviewer scoring 5 also said that he is willing to raise his score. Based on the scores, I tend to accept this work as a poster.

**Metareview: Summary, Strengths And Weaknesses:**

In this paper, the authors propose a simple but efficient transformer-based model to represent 2D/3D molecular graphs in the same framework. The proposed framework consists of a simple but effective additive mechanism to aggregate the information of 2D molecular graphs and that of 3D molecular geometry. This mechanism makes the attention map more interpretable. Experiments demonstrate the effectiveness of the proposed method.

Strengths:
(1) The idea is simple and easy to implement.
(2) Experiments are sufficient. After rebuttal, the experimental part is further enhanced.

Weaknesses:
(1) The motivation of the proposed method is not well-explained, and the authors did not provide insightful analysis for their method.


**Note From Pc:**

if the above contains the word "oral" or "spotlight" please see: "oral" presentation means -> notable-top-5% and "spotlight" means -> notable-top-25%. As stated in our emails, we are disassociating presentation type from AC recommendations